# Incorporating New Knowledge into Federated Learning: Advances, Insights, and Future Directions

**Lixu Wang**[1]*, **Yinggang Sun**[2]*, **Yang Zhao**[1]*, **Jiaqi Wu**[3], **Jiahua Dong**[4], **Ating Yin**[5], **Qinbin Li**[6], **Qingqing Ye**[7], **Dusit Niyato**[1], **Tianwei Zhang**[1], **Kwok-Yan Lam**[1], **Haining Yu**[2], **Haibo Hu**[7], **Wei Dong**[1]†

*lixu.wang@ntu.edu.sg, 23B903085@stu.hit.edu.cn, zhao0466@e.ntu.edu.sg, wujiaqi@mail.tsinghua.edu.cn, dongjiahua1995@gmail.com, atingyin@hnu.edu.cn, qinbin@hust.edu.cn, qqing.ye@polyu.edu.hk, dniyato@ntu.edu.sg, tianwei.zhang@ntu.edu.sg, kwokyan.lam@ntu.edu.sg, yuhaining@hit.edu.cn, haibo.hu@polyu.edu.hk, wei_dong@ntu.edu.sg*
[1]*Nanyang Technological University,* [2]*Harbin Institute of Technology,* [3]*Tsinghua University,* [4]*Mohamed bin Zayed University of Artificial Intelligence,* [5]*Hunan University,* [6]*Huazhong University of Science and Technology,* [7]*The Hong Kong Polytechnic University*

**Reviewed on OpenReview:** *https://openreview.net/forum?id=BWBfK3B3b7*

## Abstract

Federated Learning (FL) is a distributed learning approach that allows participants to collaboratively train machine learning models without sharing the raw data. It is rapidly developing in an era where privacy protection is increasingly valued. It is this rapid development trend, along with the continuous emergence of new demands for FL in the real world, that prompts us to focus on a very important problem: *How to Incorporate New Knowledge into Federated Learning?* The primary challenge here is to effectively and timely incorporate various new knowledge into existing FL systems and evolve these systems to reduce costs, upgrade functionalities, and facilitate sustainable development. In the meantime, established FL systems should preserve existing functionalities during the incorporation of new knowledge. In this paper, we systematically define the main sources of new knowledge in FL, including new features, tasks, models, and algorithms. For each source, we thoroughly analyze and discuss the technical approaches for incorporating new knowledge into existing FL systems and examine the impact of the form and timing of new knowledge arrival on the incorporation process. Unlike prior surveys that primarily catalogue FL techniques under a fixed system specification, we adopt a lifecycle evolution perspective and synthesize methods that enable time-varying integration of new features, tasks, models, and aggregation algorithms while preserving existing functionality. Furthermore, we comprehensively discuss the potential future directions for FL, incorporating new knowledge and considering a variety of factors, including scenario setups, security and privacy threats, and incentives.

## 1 Introduction

Federated Learning (FL), as an emerging paradigm in distributed learning, has gained significant traction and undergone substantial development over the past decade. This interest is primarily driven by its ability to enable multiple parties to collaboratively train machine learning (ML) models without sharing the raw data. Consequently, this characteristic has led to its adoption in a growing number of critical domains, such as healthcare, transportation, finance, and e-commerce, resulting in numerous real-world applications (Yang et al., 2019; Kairouz et al., 2021; Wang et al., 2021).

These practical applications demonstrate a significant demand for FL, driving its rapid development and evolution. In the field of ML, technological advancements often lead to the replacement of old models with

---

*Equal contributions. †Corresponding author.

new ones. However, given the large number of participating clients, in terms of data volume, computation overhead, and communication bandwidth, training each FL model incurs substantial costs (Kairouz et al., 2021). Furthermore, FL applications are often situated in rapidly changing environments, such as the emergence of new diseases, autonomous vehicles encountering unfamiliar roads, or the appearance of new investment trends (Yoon et al., 2021; Dong et al., 2022; Peng et al., 2020; Li et al., 2025c; Yu et al., 2023). In addition to these new concepts, there may be sensor degradation, hardware damage, product evolution, and so on, in FL deployment scenarios, such as edge computing (Wu et al., 2025d). However, current FL systems typically assume a fixed and predetermined distribution of data and tasks (Yang et al., 2019; Kairouz et al., 2021), making it challenging to deal with dynamic changes in data and tasks in real-world scenarios. These practical considerations reflect a dilemma in the future development of FL: on the one hand, there is a desire to establish new FL systems in response to emerging demands and technologies, but on the other hand, the significant investment in established FL systems makes it impractical to discard them readily. Therefore, how to achieve sustainable evolution in FL becomes a critical research direction.

Table 1: A comparative analysis of survey studies on federated learning.

| Year | Title | Difference |
|---|---|---|
| 2025 | Federated domain generalization: A survey (Li et al., 2025c) | We viewed FDG as an important tool to enable easier incorporation of new features into FL within a broader evolutionary framework. |
| | Foundational models and federated learning: survey, taxonomy, challenges and practical insights (Hatfaludi & Serban, 2025) | We focused on FL evolution through new knowledge integration rather than focusing on foundational models integration with FL. |
| | A Survey on Cluster-based Federated Learning (El-Rifai et al., 2025) | We regarded personalized FL as an effective way to prepare a task-generalizable basis for incorporating new tasks. |
| | A survey on federated fine-tuning of large language models (Wu et al., 2025f) | We talked about incorporating LLMs to enhance established FL systems rather than finetuning LLMs using FL. |
| | Federated learning design and functional models: survey (Ayeelyan et al., 2024) | We emphasize FL evolution via new knowledge integration mechanisms, rather than conventional system design analysis. |
| | Federated Continual Learning: Concepts, Challenges, and Solutions (Hamedi et al., 2025) | We reviewed studies with a much wider and more recent scope and analyzed the correlations between FCL and other new knowledge. |
| | Advances in robust federated learning: A survey with heterogeneity considerations (Chen et al., 2025a) | We focused on FL system evolution through dynamic new knowledge integration rather than addressing static heterogeneity challenges. |
| | Knowledge distillation in federated learning: a comprehensive survey (Salman et al., 2025) | We emphasize FL system evolution through dynamic new knowledge integration rather than surveying knowledge distillation in FL. |
| 2024 | Recent Advancements in Federated Learning: State of the Art, Fundamentals, Principles, IoT Applications and Future Trends (Papadopoulos et al., 2024) | We explored a more evolutionary and systematic research perspective, and defined a dynamic framework for new knowledge integration. |
| | Federated learning: Overview, strategies, applications, tools and future directions (Yurdem et al., 2024) | We explored FL dynamic evolution through new knowledge integration rather than static classification of existing strategies and applications. |
| | Recent advances on federated learning: A systematic survey (Liu et al., 2024a) | We investigated FL system evolution through new knowledge integration rather than static classification of existing FL methods. |
| | Federated continual learning via knowledge fusion: A survey (Yang et al., 2024a) | We reviewed studies with a much wider and more recent scope and analyzed the correlations between FCL and other new knowledge. |
| | A Comprehensive Survey of Federated Transfer Learning: Challenges, Methods and Applications (Guo et al., 2024b) | We reviewed techniques of dynamic FL evolution through systematic new knowledge integration. |
| 2023 | A systematic review of federated learning: Challenges, aggregation methods, and development tools (Guendouzi et al., 2023) | We emphasize FL system evolution through dynamic new knowledge integration rather than reviewing existing aggregation methods. |
| | A survey on federated learning: challenges and applications (Wen et al., 2023) | We discussed how to incorporate new demands and functionalities into the build FL systems rather than considering static FL scenarios. |

Enabling the effective evolution of FL systems requires addressing several fundamental challenges. First, these systems may continually acquire new knowledge in diverse forms. For instance, a model may need to support an increasing number of novel tasks, or even for existing tasks to generalize across broader domains. Such newly encountered knowledge is typically non-identical and independently distributed (non-IID) across clients and may arrive at different time points. Specifically, in FL, each client optimizes the model using its own data. When client data are non-IID, the updates computed on different devices are optimized for different local distributions, so they can point in inconsistent directions: an update that helps one client

may hurt others. As a result, simple averaging is no longer equivalent to taking a clean descent step on a single shared objective, which can slow convergence and reduce stability. In this setting, the FL objective at each timestep can be highly complex and potentially shift dramatically over time. Since this new knowledge often reflects transient user demands that appear sporadically, the system must be able to adapt rapidly. Second, learning new knowledge inevitably leads to the forgetting of previously acquired knowledge. This forgetting is primarily due to insufficient storage for the data of prior tasks. Furthermore, the diversity of new knowledge means that different types of old knowledge may be forgotten at different rates, complicating mitigation efforts. Finally, the dynamic evolution of FL systems also presents additional challenges, including increased computational and storage overheads, the emergence of new vulnerabilities and attack surfaces, and the breakdown of existing incentive mechanisms.

Fortunately, an increasing number of studies are starting to explore relevant problems. For instance, Sun et al. (2023) and Peng et al. (2020) achieved domain-generalizable learning to improve FL models' generalization ability of existing functionalities, and Dong et al. (2022) and Qi et al. (2022) incorporated continual learning techniques to learn new tasks. While a few existing surveys have attempted to review portions of the relevant literature, they are either outdated or lack comprehensive coverage. More critically, none of them examines these efforts through the lens of the continuous evolution of FL systems, which can inspire new thinking about FL's environmental sustainability and continual functionality iteration. A detailed comparison between these survey studies and ours is provided in Table 1. Prior surveys largely provide a snapshot taxonomy of FL methods under a fixed system specification, where the feature/task space, model architecture, and aggregation protocol are treated as predetermined, and heterogeneity is addressed within this static pipeline. Consequently, they seldom discuss the system's lifecycle questions that arise in deployments, including when new knowledge should trigger an upgrade, how the upgrade should be realized (e.g., detection, adaptation, or migration), and how to bound regressions on previously supported functionality. In this survey, we adopt a lifecycle evolution perspective and synthesize upgrade mechanisms along four evolving variables, including features, tasks, models, and aggregation algorithms, while highlighting how the arrival form and timing of new knowledge affect the incorporation process.

**Contributions:** In this paper, we present the first systematic and comprehensive survey with analysis and discussion on how FL can achieve continuous updates and development in the face of new knowledge. We define the FL system across four variables: features, tasks, models, and algorithms. Specifically, we review approaches for incorporating new features through an end-to-end workflow, including three key stages: (i) improving model generalization before new features emerge; (ii) enabling accurate detection when new features appear; (iii) facilitating model adaptation to the new features. For new tasks, we focus on enhancing the cross-task generalization ability of FL models. Given that new features and new tasks often co-occur in practice, we further explore how federated continual learning can support the continual integration of both, while maintaining strong performance on previously learned knowledge. Finally, we discuss the role of new models and algorithms as effective mechanisms for incorporating novel knowledge while preserving or even improving retention of prior capabilities. In addition, we also discuss the potential new threats and vulnerabilities and the need of new incentive mechanisms when incorporating new knowledge into FL. In summary, our major contributions include:

- We define a framework that categorizes new knowledge in FL into four types: features, tasks, models, and algorithms. For each category of new knowledge, we provide a detailed analysis and discussion on what it looks like and how to incorporate it into the current FL system in a timely and efficient manner.

- We analyze the impact of the arrival form and timing of the new knowledge on its incorporation into the FL system.

- We comprehensively discuss future research priorities for the continuous development of FL, especially with the integration of new knowledge.

## 2 Preliminaries

Federated Learning (FL) (Yang et al., 2019), as a distributed learning algorithm, enables multiple clients $\{\mathcal{C}_k\}_{k=1}^K$ to collaboratively train a machine learning model $m$ without sharing their training data $\mathcal{D}_k =$

$\{(\boldsymbol{x}_{k,i}, y_{k,i}) \sim (\mathcal{X}_k, \mathcal{Y}_k)\}_{i=1}^{N_k}$. The model $m : f_\theta \circ g_\omega$ is usually composed of a feature extractor $f_\theta$ and a task module $g_\omega$ parameterized by $\theta$ and $\omega$, respectively. In FL, clients possess their training data in a non-IID manner, i.e., the feature and task marginal distributions of different clients are not identical, $\mathcal{X}_k \neq \mathcal{X}_{k'}, \mathcal{Y}_k \neq \mathcal{Y}_{k'}$ where $k \neq k'$. In this work, we consider the horizontal FL setting, where each participant holds a distinct subset of samples with the same feature space. This differs from vertical FL, where each participant holds a subset of features for the same set of samples. A typical federated training round begins with a central server $\mathcal{S}$ distributing the latest global model $m$ to each client. Additionally, the server randomly selects a subset of clients $\{\mathcal{C}_k\}_{k=k_1}^{k_s}$ and requests them to conduct model training locally using their own data. Once local training is completed, these selected clients upload their locally trained models $m'_k$ to the server $\mathcal{S}$. The server then aggregates these models $\{m'_k\}_{k=k_1}^{k_s}$ to update the global model, and the most popular aggregation algorithm is FedAvg (McMahan et al., 2017), i.e., $m' = \frac{1}{s} \sum_{k=k_1}^{k_s} m'_k$. This process is repeated over multiple rounds until the global model converges and exhibits satisfactory performance for a given task. From the description above, we can define different FL systems $F(\mathcal{X}, \mathcal{Y}, \mathcal{M}, \mathcal{A})$ using the variables of Features $\mathcal{X} = \cup_{k=1}^K \mathcal{X}_k$; Tasks $\mathcal{Y} = \cup_{k=1}^K \mathcal{Y}_k$; Models $\mathcal{M} = \{(f_\theta, f_\alpha, \cdots) \circ g_\omega, f_\theta \circ (g_\omega, g_\beta, \cdots)\}$; Aggregation algorithms $\mathcal{A} = \{$FedAvg, FedProx (Li et al., 2020b), Moon (Li et al., 2021a), SecAgg (Bonawitz et al., 2017), CaPC (Choquette-Choo et al., 2020), $\cdots\}$.

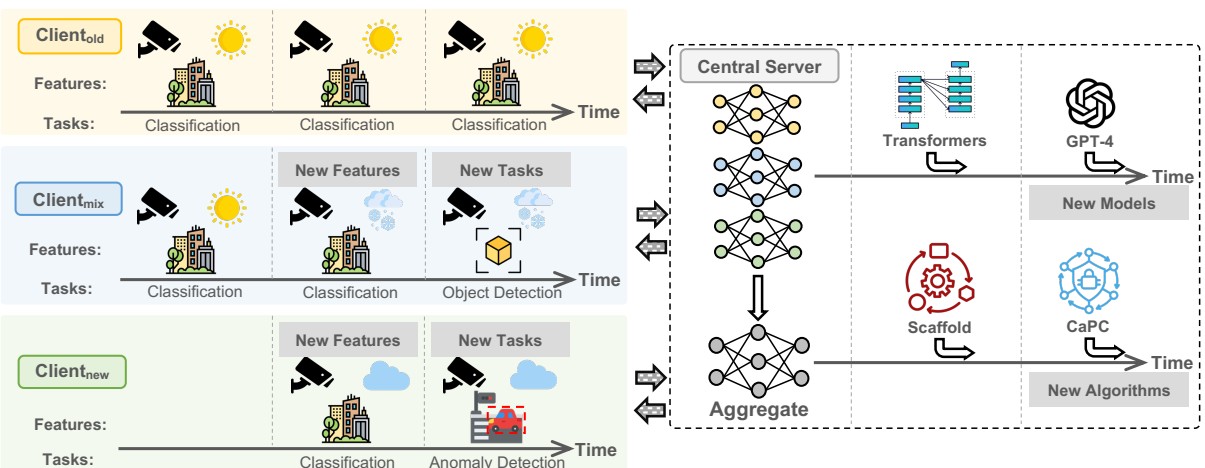

Figure 1: Overview of an FL system with new knowledge from different sources. Different types of clients encounter new features and tasks over time, which reflect new demands for FL systems. From a global perspective, new advanced models with better architecture (Transformers) and larger sizes (GPT-4) are also needed to incorporate for enhancing existing functionalities or incorporating new demands. Furthermore, new algorithms with improved performance (Scaffold) and security guarantees (CaPC) should be continuously incorporated into established FL as well.

## 2.1 Federated Learning with New Knowledge

Starting from the moment of the first training round, we denote the existing period of an FL system $F(\mathcal{X}, \mathcal{Y}, \mathcal{M}, \mathcal{A})$ until the present time $t$ as as a sequence of time stamps $[1, 2, \cdots, t, \cdots]$, then the new knowledge is defined as a particular variable at a particular timestamp that is unseen previously. If we take the feature variable as an example, we have $\mathcal{X}^t \not\subseteq \cup_{i=1}^{t-1} \mathcal{X}^i$. To facilitate the coming of more new knowledge, we do not restrict clients' participation, which means clients are free to join or leave the FL system at any time. Under this framework, we categorize the FL clients into three groups: $\mathcal{C}_{\text{old}} = \{\mathcal{C}_k\}_{k=k_{\text{old},1}}^{k_{\text{old},n_{\text{old}}}}$ corresponds to clients that possess only old knowledge; $\mathcal{C}_{\text{new}} = \{\mathcal{C}_k\}_{k=k_{\text{new},1}}^{k_{\text{new},n_{\text{new}}}}$ corresponds to clients that only possess new knowledge; $\mathcal{C}_{\text{mix}} = \{\mathcal{C}_k\}_{k=k_{\text{mix},1}}^{k_{\text{mix},n_{\text{mix}}}}$ corresponds to clients that possess both old and new knowledge. Note that the category belonging of a particular client $\mathcal{C}_k$ is changing over time; thus, these three client groups $\mathcal{C}_{\text{old}}, \mathcal{C}_{\text{new}},$ and $\mathcal{C}_{\text{mix}}$ are also changing dynamically. Overall, regardless of which variable the new knowledge belongs to,

when it arrives, or who introduces it, we need to incorporate the new knowledge promptly. Incorporating such new knowledge is crucial because the arrival of new knowledge essentially signifies clients' new demands for enhancing the performance and functionality of the current FL system. We visualize an FL system with various kinds of new knowledge in Figure 1. Within such setups, we primarily focus on cross-device FL in the following sections, as it offers greater flexibility for client participation and withdrawal compared to cross-silo FL. This setting typically involves a larger number of parties, components, and concepts, all of which contribute to higher system dynamics. In addition, cross-device FL systems are often resource-constrained. Overall, incorporating new knowledge into cross-device FL systems is more needed than cross-silo FL.

## 2.2 Security Setups of FL with New Knowledge

A trusted setting is assumed when incorporating new knowledge into an FL system, i.e., (i) all participants including both the central server $\mathcal{S}$ and clients $\{\mathcal{C}_k\}_{k=1}^K$ are trusted not to eavesdrop private information or launch active attacks at any time; (ii) all carriers of new knowledge including features $\mathcal{X}$, tasks $\mathcal{Y}$, models $\mathcal{M}$, and algorithms $\mathcal{A}$ do not intend to disclose private information or contain active attacks.

The reason why we consider such trusted settings is that we aim to discuss the technical advances of FL with new knowledge in the main context, and we found that all existing related works assume such trusted scenarios. However, this does not mean that the security and safety research is unimportant; instead, we regard it as highly important and discuss new threats in both semi-honest and malicious settings as well as the potential solutions in Section 8.2.

## 3 Incorporate New Features

When FL models are deployed on new devices with unseen features, performance often degrades due to the discrepancy between learned features (source domain) and new features (target domain) (Shenaj et al., 2023). Therefore, it is important to improve the generalization ability of FL systems to unseen new features. Specifically, we consider that Federated Domain Generalization (FDG) can equip the FL system with a certain degree of generalization ability on unseen features. Federated Out-of-Distribution Detection (FODD) can detect the arrival of new features and then determine whether to incorporate them. After that, Federated Domain Adaptation (FDA) conducts adaptive learning on unlabeled data with new features, achieving the incorporation of new features.

## 3.1 Federated Domain Generalization

FDG aims to train a domain-generalizable global model $m$ that can generalize well across different distributions. To provide a structured overview of the technical landscape, we summarize and compare representative FDG approaches in Table 2 in terms of their timing support, overhead, and utility. Such domain-generalizable learning is usually achieved by extracting domain-shared semantics across local clients with old knowledge $\mathcal{C}_{\text{old}}, \mathcal{C}_{\text{mix}}$ (source domains). In this way, model $m$ can extract similar semantics from the unseen new features $\mathcal{X}_{\text{new}}$. Here, we assume that $\mathcal{X}_{\text{new}}$ is relevant but distinct from $\cup_{k=1}^K \mathcal{X}_k$ in terms of the same task $\mathcal{Y}$.

### 3.1.1 Methods

**Distribution Alignment.** Several FDG methods (Li et al., 2025c) employ feature alignment techniques to minimize domain discrepancy. Generally, they achieve this by optimizing with distribution alignment regularizers (Xu et al., 2021), such as contrastive loss, adversarial loss, and maximum mean discrepancy (MMD). For example, Xu et al. (2023) proposed a negative-free contrastive loss at the logit level to minimize the distribution gap between the original sample and its hallucinated counterparts. Building on this work, Gupta et al. (2025) enhanced model generalization through domain invariant feature perturbation alignment. Regarding the adversarial loss, Wang et al. (2022c) used adversarial learning for multi-client feature alignment. Similarly, Zhang et al. (2023) introduced another generalization adjustment model via dynamically calibrating aggregation weights with an adversarial objective. As for the MMD aspect, Tian et al. (2023) used MMD for gradient alignment, encouraging aggregated gradients to unify information from multiple domains. Additional gradient-based distributional alignment approaches have emerged. Wei & Han

Table 2: Comparison of approaches in Federated Domain Generalization. While **distribution alignment** strategies typically incur higher costs and possess limited scenario support, **data augmentation** and **optimization enhancement** approaches effectively address these limitations by offering improved flexibility and superior efficiency, respectively.

| Strategy | Methods | Timing Support | | Period Support | | Synchronism | | Overhead | | | Utility |
|---|---|---|---|---|---|---|---|---|---|---|---|
| | | Abrupt | Graduate | Short | Long | Sync | Async | Computation | Communication | Storage | Performance |
| Distribution Alignment | FADH (Xu et al., 2023) | ✓ | ✓ | ✓ | ✓ | ✓ | ✓ | Medium | Low | Low | Near SOTA |
| | FedAlign$_{DG}$* (Gupta et al., 2025) | ✓ | ✓ | ✓ | ✓ | ✓ | ✓ | Medium | Low | Low | SOTA |
| | FADGN (Wang et al., 2022c) | ✓ | × | × | ✓ | ✓ | × | High | Medium | Low | Behind SOTA |
| | FedADG (Zhang et al., 2023) | ✓ | × | × | ✓ | ✓ | × | High | Medium | Low | Near SOTA |
| | PCDG (Tian et al., 2023) | ✓ | × | × | ✓ | ✓ | × | Medium | High | Low | Near SOTA |
| | FedDM (Sun et al., 2023) | ✓ | ✓ | × | ✓ | ✓ | × | High | High | Medium | Near SOTA |
| | MCGDM (Wei & Han, 2024) | ✓ | × | × | ✓ | ✓ | × | High | High | Low | Near SOTA |
| | FedU (Pourpanah et al., 2025) | ✓ | ✓ | ✓ | ✓ | ✓ | ✓ | Medium | Low | Low | SOTA |
| | FedGM (Nguyen et al., 2025) | ✓ | × | × | ✓ | ✓ | × | Medium | High | Low | SOTA |
| Data Augmentation | FedDG (Liu et al., 2021) | ✓ | ✓ | ✓ | ✓ | ✓ | ✓ | Low | Low | Low | Behind SOTA |
| | FedCST (Chen et al., 2023b) | ✓ | ✓ | ✓ | ✓ | ✓ | ✓ | Medium | Medium | Medium | Near SOTA |
| | FISC (Nguyen et al., 2024a) | ✓ | ✓ | ✓ | ✓ | ✓ | ✓ | Medium | Medium | Medium | SOTA |
| | FedGCA (Liu et al., 2024b) | ✓ | ✓ | × | ✓ | ✓ | ✓ | High | Low | Low | Near SOTA |
| | CAL-ZSA (Yang et al., 2023c) | ✓ | ✓ | ✓ | ✓ | ✓ | ✓ | Low | Low | Low | Near SOTA |
| | FedCCRL (Wang et al., 2024c) | ✓ | ✓ | ✓ | ✓ | ✓ | ✓ | Medium | Medium | Low | SOTA |
| Optimization Enhancement | FedLS (Soltany et al., 2024) | ✓ | ✓ | ✓ | ✓ | ✓ | ✓ | Low | Low | Low | Near SOTA |
| | BiPFed (Lai et al., 2024) | ✓ | ✓ | ✓ | ✓ | ✓ | ✓ | Low | Low | Low | Near SOTA |
| | FedSemiDG (Deng et al., 2025) | ✓ | ✓ | ✓ | ✓ | ✓ | ✓ | Low | Low | Low | SOTA |

(2024) proposed reducing distribution differences through intra-domain and inter-domain gradient matching. Nguyen et al. (2025) regarded local gradients as the representation of specific domains and maximized the inner product of gradients to find invariant gradient directions across all domains. Similarly, Ye et al. (2025) proposed the GAFedDG framework, which combines intra-domain gradient alignment between raw and augmented signals with inter-domain gradient alignment across domain classifiers to achieve effective distribution alignment.

**Data Augmentation.** To improve the model's generalization to unseen target domains, some FDG methods use data augmentation techniques to diversify training data distributions (Liu et al., 2023). Common augmentation techniques include domain randomization, domain generation, and domain mixup. Liu et al. (2021) used domain randomization techniques like random rotation, scaling, and flipping to address the FDG problem. However, as research progressed, the limitations of simple domain randomization in tackling complex domain shift issues became clear. Chen et al. (2023b) introduced a cross-client style transfer method, where local clients define their styles and share them with a central server. Building on this concept, Nguyen et al. (2024a) proposed the FISC method, which refines cross-domain learning by extracting an interpolative style from local client styles and using contrastive learning. Simultaneously, efforts were made to enhance data diversity through domain generation. Liu et al. (2024b) increased data diversity by generating samples with diverse domain styles using a style-complement module. For domain mixup, Yang et al. (2023c) proposed mixing local and global feature statistics by randomly interpolating instance and global statistics. Wang et al. (2024c) followed this by boosting local domain diversity through cross-client domain transfer and domain-invariant feature perturbation.

**Optimization Enhancement.** Many FDG methods introduce regularization terms to prevent the model from overfitting domain-specific features. For instance, Soltany et al. (2024) proposed client-side label smoothing to reduce the model's reliance on domain-specific features, enhancing generalization across diverse domains. To further ensure stable and robust global model updates against data heterogeneity, Lai et al. (2024) proposed a distance-based weighted aggregation combined with parameter moving averaging. Building on these advancements, Deng et al. (2025) developed a generalization-aware aggregation method, dynamically adjusting aggregation weights based on each client model's cross-domain performance.

---

*We add the subscript "$_{DG}$" to FedAlign (Gupta et al., 2025) to differentiate from the studies (Ravi & Shomorony (2024), Ma et al. (2024b)) with the same name. Ravi & Shomorony (2024) introduced FedAlign to select and unify non-priority clients to align with the priority clients. Ma et al. (2024b) proposed FedAlign to achieve alignment across different modalities in FL.

Table 3: Commonly used experimental datasets in incorporating new knowledge into FL systems.

| Dataset | Sample | Task Description |
|---|---|---|
| PACS (Li et al., 2017) | 9,991 | Images of daily objects with 4 artistic styles, covering 7 classes. |
| DomainNet (Peng et al., 2019) | 600,000 | Images of daily objects collected from 6 different domains with 345 classes. |
| Digits-DG (Zhou et al., 2020) | 2,400 | Images of handwritten digits with 4 domains, covering 40 classes. |
| VLCS (Shahtalebi et al., 2021) | 10,729 | Images of common objects from 4 domains, covering 5 classes. |
| TerraIncognita (Beery et al., 2018) | 24,000 | Wildlife camera images from 20 locations, covering 10 classes. |
| Camelyon17 (Bai et al., 2023) | 410,359 | Lymph node slice image data, covering 5 domains and 2 classes. |
| FEMNIST (Caldas et al., 2018) | 800,000 | Images of handwritten digits and characters across 3 domains and 62 classes. |
| CIFAR-10 (Krizhevsky et al., 2009) | 60,000 | A popular small-scale natural image classification dataset with 10 classes. |
| CIFAR-100 (Krizhevsky et al., 2009) | 60,000 | A dataset similar to CIFAR-10 but with 100 classes. |
| CIFAR-10-C (Hendrycks & Dietterich, 2019) | 10,000 | CIFAR-10 with damage processing from 19 domains, covering 10 classes. |
| VisDA2017 (Peng et al., 2017) | 280,000 | Images from synthetic rendering to real shooting, covering 12 classes. |
| ImageCLEF-DA (Zhang et al., 2019) | 1,800 | Images from 3 collection scenarios, covering 12 classes. |
| Office-31 (Saenko et al., 2010) | 4,110 | Office supply images from 3 photography scenarios, covering 31 classes. |
| Office-Home (Venkateswara et al., 2017) | 15,588 | Office supply images from 4 domains, covering 65 classes. |
| Office-Caltech10 (Zhang & Davison, 2020) | 2,500 | Office supply images from 4 domains, covering 13 classes. |
| Digit-Five (Saito et al., 2018) | 700,000 | Handwritten digit images from 5 domains, covering 10 classes. |
| Tiny ImageNet (Le & Yang, 2015) | 100,000 | Images of daily objects and a subset of ImageNet, covering 200 classes. |
| Omniglot (Song et al., 2022) | 1,623 | Images of handwritten characters from 50 different alphabets, with 50 classes. |
| Shakespeare (Caldas et al., 2018) | 4,226,158 | Text data of all works by Shakespeare. |
| CelebA (Liu et al., 2015) | 202,599 | Face images with detailed attribute annotations. |
| Sent140 (Caldas et al., 2018) | 1,600,000 | Texts collected from Twitter for sentiment analysis, with 2 classes. |
| SVHN (Netzer et al., 2011) | 600,000 | Images of real-world house number digits, with 10 classes. |
| COVID-FL (Yan et al., 2023) | 20,000 | Medical images for COVID-19 FL research, covering 4 classes. |
| Kinetics 400 (Kay et al., 2017) | 306,245 | Videos of real-world human actions, covering 400 classes. |
| UCF101 (Soomro et al., 2012) | 13,320 | Video clips of human actions in different scenarios, covering 5 classes. |

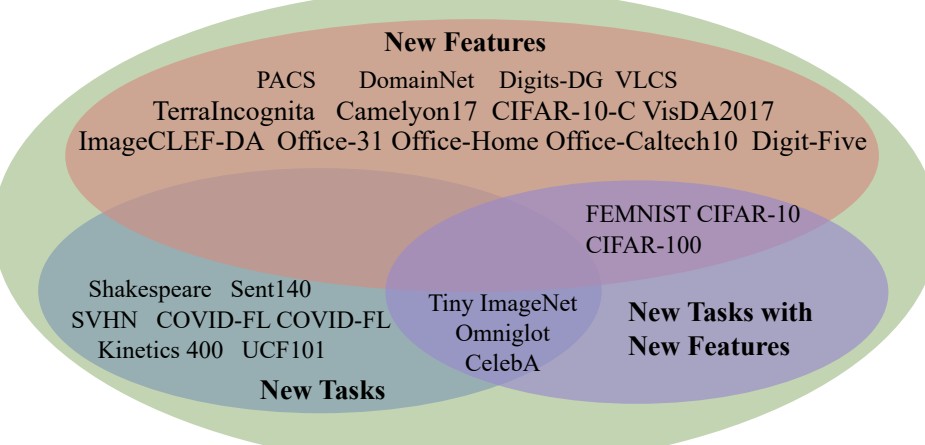

Figure 2: The mapping relationship between the commonly used datasets and different new knowledge types.

### 3.1.2 Evaluation

**Datasets.** The commonly used experimental datasets for FDG, as shown in Table 3 include PACS, DomainNet, Digits-DG, VLCS, TerraIncognita, Camelyon17, Office-31, Office-Home, and Office-Caltech10.

**Setup.** The most common FDG configuration is the single-client-single-domain setup, where each client owns data from one source domain. To better mimic real-world scenarios where a client may have data from multiple sources, some studies explore intra-client domain heterogeneity setups. For example, Stable-FedADG (Zhao et al., 2025a) allowed clients to hold unbalanced data mixtures from multiple source domains or include dominant and secondary domains in a "mixed" setup. Additionally, this setup (Bai et al., 2023) introduced a parameter $\lambda$ to control inter-client domain heterogeneity, highlighting the impact of different data distributions and heterogeneity levels.

Table 4: Comparison of approaches in Federated Out-of-Distribution Detection. Notably, **FedDD** distinguishes itself with minimal overhead across all dimensions, while **FedOE** achieves state-of-the-art performance with moderate resource consumption.

| Methods | Timing Support | | Period Support | | Synchronism | | Overhead | | | Utility |
|---|---|---|---|---|---|---|---|---|---|---|
| | Abrupt | Graduate | Short | Long | Sync | Async | Computation | Communication | Storage | Performance |
| FeAnD (Nardi et al., 2022) | ✓ | ✓ | × | ✓ | ✓ | ✓ | Medium | Medium | Medium | Behind SOTA |
| FedLSTM-AE (Mohammadi et al., 2023) | ✓ | ✓ | × | ✓ | ✓ | × | Medium | Medium | Medium | Near SOTA |
| FedOoD (Yu et al., 2023) | ✓ | × | × | ✓ | ✓ | × | High | Medium | Low | Near SOTA |
| OT-FODD (He et al., 2025a) | ✓ | × | × | ✓ | ✓ | × | High | Medium | Low | SOTA |
| FedOE (Jeong & Choi, 2025) | ✓ | ✓ | ✓ | ✓ | ✓ | × | Medium | Medium | Low | SOTA |
| FedDD (Rahimli et al., 2024) | ✓ | ✓ | ✓ | ✓ | ✓ | ✓ | Low | Low | Low | Near SOTA |
| FOOGD (Liao et al., 2024) | ✓ | ✓ | × | ✓ | ✓ | × | Medium | Medium | Low | SOTA |
| FedFISS (Dong et al., 2023) | ✓ | ✓ | ✓ | ✓ | ✓ | ✓ | Low | Medium | Low | Near SOTA |
| FedCNCL (Wang et al., 2025a) | ✓ | ✓ | × | ✓ | ✓ | × | High | Medium | Medium | Near SOTA |

**Metric.** Four metrics measure how well an FL model generalizes to unseen target domains: accuracy on unseen target domains, average accuracy across all domains, worst domain accuracy, and accuracy difference across domains.

### 3.1.3 Real-world Applications

FedBM (Zhu et al., 2025b) used FDG principles to address medical data heterogeneity, achieving accurate and robust medical image analysis and diagnosis by eliminating local learning biases. DGSSL (Liu et al., 2025a) proposed semi-supervised domain generalization learning, combining autoregressive discriminators and reconstruction tasks to achieve robust and accurate recognition by addressing cross-individual domain shifts in human activity recognition. DG-FaultDiag (Zhao et al., 2024a) addressed multi-source data privacy and distribution differences in industrial scenarios by integrating domain generalization techniques with FL, enabling robust fault diagnosis for different devices and conditions.

### 3.2 Federated Out-of-Distribution Detection

FODD was designed to detect new features across local clients (typically $\mathcal{C}_{\text{mix}}$ and $\mathcal{C}_{\text{new}}$) using a binary classifier $g_b$ placed after the global feature extractor $f_\theta$. To provide a structured overview, we summarize and compare representative FODD approaches in Table 4 regarding their detection strategies, overhead, and utility. Once new feature data is detected, it should be incorporated into the current FL system. Therefore, $g_b$ should be trained to determine if a sample $x$ comes from a previously unseen feature distribution $x \sim \cup_{i=1}^{t-1}\mathcal{X}_i$. The following section reviews relevant studies.

### 3.2.1 Methods

Several approaches have been used to achieve FODD. Reconstruction prediction is one approach. Nardi et al. (2022) proposed a decentralized anomaly detection framework. Clients are clustered into communities based on the similarity of their representations, followed by collaborative training of a federated autoencoder. Similarly, Mohammadi et al. (2023) used a long short-term memory mechanism within a modified autoencoder to detect outliers in FL systems. Another effective approach is synthesizing potential out-of-distribution (OOD) data and differentiating it from in-distribution (ID) data. For instance, Yu et al. (2023) proposed training category-specific generators to create OOD samples for detector training. He et al. (2025a) proposed generating pseudo-OOD samples via the Wasserstein distance, creating virtual outlier samples from known samples to address the scarcity of OOD data in federated environments. Instead of generating OOD data, Jeong & Choi (2025) assumed the server stores some outlier samples while clients keep minimal normal data. This approach ensures efficient outlier detection through internal feature separation and background collaboration. Feature-based detection represents a third approach, which relies on feature extractors to characterize ID data and employs lightweight detection mechanisms. Rahimli et al. (2024) used Fisher's Exact Test to identify concept and data drift. Liao et al. (2024) developed the FOOGD framework, modeling client data distributions with an OOD scoring mechanism and achieving end-to-end FODD by integrating semantic

shift detection with a covariate shift generalization module. Dong et al. (2023) introduced a task transition monitor using classification entropy of client data representations to identify OOD tasks, enabling real-time adaptation to evolving data patterns. Wang et al. (2025a) developed a hierarchical clustering mechanism on data representations to discover and learn previously unobserved features.

### 3.2.2 Evaluation

**Datasets.** The commonly used datasets for FODD, as shown in Table 3, include FEMNIST, CIFAR-10/CIFAR-100, CIFAR-10-C, PACS, Digit-Five, and Office-Home.

**Setup.** The evaluation setup of FODD primarily focuses on two scenarios: unseen class detection and feature shift detection. First, unlike conventional label shift, which only causes an imbalance in the number of categories, FODD simulates scenarios where clients detect previously unseen categories. This is an extreme form of label heterogeneity (Liao et al., 2024), typically constructed using a Dirichlet distribution with a small parameter $\alpha$. It requires the model to determine whether an unknown sample comes from known categories of other clients or is entirely new for all clients. Second, FODD is evaluated in scenarios with feature shifts, where clients share labels but the data features come from different distributions. This is demonstrated in various benchmark tests, such as the domain migration experiments of FOSTER (Yu et al., 2023) on DomainNet and the HDR-FL benchmark used by FedPD (Yang et al., 2023a).

**Metric.** Five metrics are used to evaluate FODD performance: the area under the receiver operating characteristic curve, the area under the precision-recall curve, the false positive rate at 95% true positive rate, the true negative rate at 95% true positive rate, and the F1 Score.

### 3.2.3 Real-world Applications

Li et al. (2025a) proposed a secure and backpropagation-free FODD approach, useful for local OOD identification and data quality control in FL, as well as detecting OOD threats in video analysis. Zamzmi et al. (2024) suggested using statistical process control to monitor changes in the statistical characteristics of medical image data streams, enabling early identification of potential data drift and enhancing FODD response capabilities in radiology. Zhang et al. (2024f) introduced a deep ensemble method guided by maximum consistency and minimum similarity, significantly improving the model's uncertainty estimation and OOD sample discrimination.

## 3.3 Federated Domain Adaptation

FDA aims to collaboratively adapt the global model to data with new features detected by FODD. We assume there is a significant domain discrepancy between the old and new features, $d(\mathcal{X}_{\text{new}}, \mathcal{X}_k) \gg \max d(\mathcal{X}_k, \mathcal{X}_{k'})$ where $d(\cdot)$ denotes a function measuring distribution discrepancy. As most data with new features appears without labels, FDA should address the domain discrepancy between the labeled source data with old features and the unlabeled target data with new features. To facilitate a clear comparison, we summarize representative FDA approaches in Table 5, evaluating their support for different learning scenarios (timing, periodicity, and synchronism), associated overheads, and overall utility.

### 3.3.1 Methods

**Adversarial Training.** The difference in feature distributions between domains is a major obstacle to effective knowledge transfer. Adversarial training aligns inter-domain features by training discriminators and feature extractors adversarially. Peng et al. (2020) were the first to extend adversarial adaptive training to FL, aiming to align learning representations of different nodes with the target node's data distribution. They enhanced knowledge transfer by combining dynamic attention mechanisms and feature disentanglement. More studies have enhanced this adversarial framework in various ways. First, Wu et al. (2024b) proposed decoupling features using mutual information theory before adversarial training. This approach ensures that minimizing distribution differences is not affected by domain-specific features. Additionally, Chi et al. (2024) designed multiple tightly coupled joint classifiers for adversarial training to promote alignment of the joint distribution of features and classes. Zhang & Li (2022) combined federated deep adversarial networks

Table 5: Comparison of approaches in Federated Domain Adaptation. While **adversarial training** strategies achieve state-of-the-art performance, they generally suffer from high computational costs and limited scenario support. In contrast, **pseudo-label learning** and **distribution alignment** approaches offer superior flexibility in timing and synchronism with significantly reduced overheads.

| Strategy | Methods | Timing Support | | Period Support | | Synchronism | | Overhead | | | Utility |
|---|---|---|---|---|---|---|---|---|---|---|---|
| | | Abrupt | Graduate | Short | Long | Sync | Async | Computation | Communication | Storage | Performance |
| Adversarial Training | FADA (Peng et al., 2020) | ✓ | × | × | ✓ | ✓ | × | High | Medium | Low | Behind SOTA |
| | FedFDDA (Wu et al., 2024b) | ✓ | × | × | ✓ | ✓ | × | High | Medium | Low | SOTA |
| | FMDADA (Chi et al., 2024) | ✓ | × | × | ✓ | ✓ | × | High | Medium | Low | SOTA |
| | FTLAN (Zhang & Li, 2022) | ✓ | × | × | ✓ | ✓ | × | High | Medium | Low | Near SOTA |
| | FedDASK Gong et al. (2024a) | ✓ | × | × | ✓ | ✓ | × | High | Medium | Low | SOTA |
| | FMTDA (Yao et al., 2022) | ✓ | × | × | ✓ | ✓ | × | High | Medium | Low | Near SOTA |
| | GM-FDA (Zeng et al., 2022) | ✓ | × | × | ✓ | ✓ | × | High | Medium | Low | Near SOTA |
| Pseudo-label Learning | FedUDA (Zhuang et al., 2022) | ✓ | ✓ | ✓ | ✓ | ✓ | ✓ | Medium | Medium | Low | Near SOTA |
| | FMSDAA (Zhao et al., 2023) | ✓ | ✓ | ✓ | ✓ | ✓ | ✓ | Medium | Medium | Low | Near SOTA |
| | COPA (Wu & Gong, 2021) | ✓ | ✓ | ✓ | ✓ | ✓ | ✓ | Medium | Medium | Low | Near SOTA |
| | FedFTL (Li et al., 2024d) | ✓ | ✓ | ✓ | ✓ | ✓ | ✓ | Medium | Medium | Low | SOTA |
| | FedWCA (Mori et al., 2024) | ✓ | ✓ | ✓ | ✓ | ✓ | ✓ | Medium | Medium | Low | SOTA |
| Distribution Alignment | BFTL (Wang et al., 2024b) | ✓ | ✓ | ✓ | ✓ | ✓ | ✓ | Medium | Medium | Low | SOTA |
| | HFSDA (Rizzoli et al., 2024) | ✓ | ✓ | ✓ | ✓ | ✓ | ✓ | Medium | Medium | Low | SOTA |
| | FedRF (Wang et al., 2024d) | ✓ | ✓ | ✓ | ✓ | ✓ | ✓ | Medium | Low | Low | SOTA |

at both the image and pixel levels to comprehensively reduce the feature gap between source and target domains. Gong et al. (2024a) introduced cross-domain semantic similarity measurements to conventional domain-level adversarial training, accelerating convergence and improving the quality of aligned features. Moreover, to address the single-source multi-target scenario, Yao et al. (2022) proposed a federated dual adaptation strategy, achieving cross-domain feature alignment through local classifier adaptation on the client side and Gaussian mixture model weighted mixup on the server side. Similarly, Zeng et al. (2022) reduced domain shift in multi-site brain imaging data through adversarial domain adaptive pre-training.

**Pseudo-label Learning.** The lack of annotations for target domain data limits model adaptation, so a natural approach is to generate pseudo-labels to include them in the learning process. Zhuang et al. (2022) proposed the FedFR method, introducing distance constraints into the clustering algorithm to improve pseudo-label accuracy. In multi-source domain adaptation scenarios, Zhao et al. (2023) enhanced pseudo-label accuracy through a joint voting scheme. Similarly, Wu & Gong (2021) used a prediction consistency mechanism to generate more accurate pseudo-labels and compute aggregation weights, facilitating effective multi-source domain adaptation. Considering not only domain shift but also the more complex issue of class shift, Li et al. (2024d) proposed generating pseudo-labels through global clustering and local semantic consistency clustering. For the challenging federated source-free domain adaptation task, where source data is inaccessible, Mori et al. (2024) proposed a weighted clustering aggregation method. This method first clusters clients to obtain domain-specific global models and then performs local adaptive fine-tuning on clients using the generated pseudo-labels.

**Distribution Alignment.** Reducing the distribution difference between the source and target domains is a straightforward approach to domain adaptation. A common method is to use MMD to measure and minimize the domain gap. Wang et al. (2024d) proposed aligning these domains by minimizing the MMD distance using random features and employing low-rank approximation techniques to compress exchanged information, enhancing communication efficiency. Unlike aligning on the client side locally, Wang et al. (2024b) used MMD as a dynamic weight to balance the contributions of different local models. Another major category of methods aligns feature distributions through prototype learning. For instance, Rizzoli et al. (2024) proposed to expand the geometric space of prototype learning from the traditional Euclidean space to the hyperbolic space, which is more suitable for hierarchical data.

### 3.3.2 Evaluation

**Datasets.** The commonly used experimental datasets for FDA, as shown in Table 3, include VisDA2017, ImageCLEF-DA, Office-31, Office-Home, Office-Caltech10, Digit-Five, PACS, DomainNet, and VLCS.

**Setup.** Most FDA studies employ a setup with multiple source-domain clients and a single target-domain client. Source-domain clients have labeled data from one or more source domains, while the target-domain client has data from a different target domain, which may be minimally labeled or completely unlabeled.

**Metric.** Two metrics are commonly used: average accuracy and F1 Score on the target domain test data.

### 3.3.3 Real-world Applications

In the medical field, FedWeight (Zhu et al., 2025a) applied a distribution quantization method based on density estimation to accurately capture patient data characteristics at each medical institution, effectively addressing covariate shift in electronic health record processing. In autonomous driving, FFreeDA (Mori et al., 2025) improved semantic segmentation performance under adverse weather conditions. In edge computing, Li et al. (2024b) adapted WiFi-based human activity recognition models to new users or environments by aggregating diverse hypotheses from different domains, significantly reducing reliance on labeled target domain data. Wu et al. (2025e) developed an efficient anomaly object detection tool that can generalize to different scenarios.

## 3.4 Future Work

We envision that future FL will evolve into a highly integrated and automated end-to-end framework to more efficiently incorporate new features. Ideally, when an FL system encounters features outside its training distribution, it should first exhibit robust OOD generalization. Following this, the framework must accurately detect the arrival of these new features. Once detected, the system should automatically trigger a data collection process. When the new data reaches a sufficient scale or meets predefined criteria, the system will seamlessly initiate an adaptation process to integrate the new knowledge into the existing model. Within this overarching vision, future work on each component can focus on the following aspects:

**FDG.** Future research could pursue more fundamental theoretical and paradigm innovations based on existing work. Theoretically, while deriving stricter generalization bounds remains important (Wang et al., 2025c), a greater challenge is to establish a framework that unifies the trade-offs among generalization, privacy, and efficiency, guiding algorithm design under extreme heterogeneity. In terms of model paradigms, although studies have begun adapting large models to federated scenarios using techniques like prompt learning (Gong et al., 2024b), future research could explore constructing a shared adapter library with diverse domain knowledge. Clients can flexibly use these modules based on dynamic needs at minimal cost. Additionally, regarding learning paradigms, beyond exploring unsupervised FDG to reduce reliance on labels (Yan & Guo, 2024), designing proxy tasks that require models to distinguish data distributions across clients may enable learning of essential domain-invariant features.

**FODD.** Current methods mainly focus on enhancing OOD generalization through model intervention or knowledge distillation (Qi et al., 2025b), which is essentially passive. Future research should shift to proactive and predictable OOD response mechanisms. By exploring new perspectives like data generation and feature space reconstruction, we can systematically enhance the OOD generalization ability of FL systems when facing data distribution shifts. New FL architectures can enable autonomous learning and dynamic adjustments for OOD problems, e.g., some (virtual) clients can be dedicated to monitoring and detecting new data distributions by comparing aggregated global models over time.

**FDA.** Current research is largely empirically driven and lacks rigorous theoretical guidance. While some work provides preliminary theoretical analysis (Feng et al., 2024b), future breakthroughs lie in understanding the correlation between the generalization of adapted models and heterogeneity in FL. In terms of efficiency, although one-shot methods reduce communication overhead (Abedi et al., 2025; Zhang et al., 2024c), they often lack flexibility. Future work could explore novel mechanisms to minimize the extra communication overhead incurred when transmitting domain-specific information. Additionally, customizing recent test-

time adaptation approaches to FL setups is worth studying. Such customization should consider adjusting test-time adaptation according to data heterogeneity while preserving client privacy.

# 4 Incorporate New Tasks

In the dynamic field of FL, traditional systems face significant challenges when addressing the diverse and continuously emerging tasks from clients. The introduction of new functionalities, similar to new tasks with comparable features, is common. To effectively respond to these evolving requirements, it is crucial to enhance the cross-task generalization capabilities of FL models. This need directs our focus towards advanced methodologies in FL, specifically task-personalized FL and self-supervised FL.

## 4.1 Task-Personalized Federated Learning

Task-Personalized FL (TPFL) is essential to satisfy the diverse and specific requirements of clients within FL systems. This approach deals with the inherent diversity in feature and task distributions among different clients. Each client $\mathcal{C}_k$ adapts the global model $m : f_\theta \circ g_\omega$ to better align with its local data characteristics and specific needs. Initially, the central server $\mathcal{S}$ distributes the global model to each client. The clients then use their unique datasets $\mathcal{D}_k$ to locally train models $m'_k$, optimizing a personalized objective function $\mathcal{L}_k [m'_k(\mathcal{X}_k), \mathcal{Y}_k]$. These models are aggregated by the server using FL algorithms from set $\mathcal{A}$, resulting in an updated global model $m'$ with better cross-task generalization ability due to this multi-task training process. When new tasks emerge in the future, the adaptation can be easily achieved by a little transfer learning from the latest global model. To facilitate a comprehensive understanding, we summarize representative TPFL approaches in Table 6, comparing their support for learning scenarios, computational and communication overheads, and overall utility.

Table 6: Comparison of approaches in Task-Personalized Federated Learning. While **client grouping** methods typically lack support for asynchronous settings and impose strict scenario constraints, **regularized optimization** strategies demonstrate superior flexibility, offering broader applicability across diverse timing and synchronization requirements.

| Strategy | Methods | Timing Support | | Period Support | | Synchronism | | Overhead | | | Utility |
|---|---|---|---|---|---|---|---|---|---|---|---|
| | | Abrupt | Graduate | Short | Long | Sync | Async | Computation | Communication | Storage | Performance |
| Regularized Optimization | pFedMe (T Dinh et al., 2020) | ✓ | ✓ | ✓ | ✓ | ✓ | ✓ | High | Medium | Low | Behind SOTA |
| | FedProto (Tan et al., 2022) | ✓ | ✓ | ✓ | ✓ | ✓ | ✗ | Low | Low | Low | Behind SOTA |
| | FL-ERS (Li et al., 2024c) | ✓ | ✓ | ✓ | ✓ | ✓ | ✓ | Low | Low | Low | SOTA |
| | FedAS (Yang et al., 2024b) | ✓ | ✓ | ✗ | ✓ | ✓ | ✗ | High | Medium | Low | SOTA |
| | FedRE (Yao et al., 2026) | ✓ | ✓ | ✓ | ✓ | ✓ | ✗ | Low | Low | Low | SOTA |
| Client Grouping | FedCS (Han et al., 2023) | ✓ | ✗ | ✗ | ✓ | ✓ | ✗ | Medium | Medium | Low | Near SOTA |
| | MiniPFL (Fan et al., 2024) | ✓ | ✓ | ✓ | ✓ | ✓ | ✗ | Medium | Medium | Low | SOTA |
| | CA-PFL (Zhao et al., 2024b) | ✓ | ✓ | ✗ | ✓ | ✓ | ✗ | High | Medium | Low | SOTA |
| | APFL-OS (Ge et al., 2024) | ✓ | ✗ | ✓ | ✓ | ✓ | ✗ | Low | Low | Low | SOTA |

### 4.1.1 Methods

**Regularized Optimization.** Regularized optimization in TPFL effectively addresses data and model heterogeneity by customizing task-specific modules, while a shared global feature extractor facilitates knowledge aggregation across tasks. A common approach involves decoupling updates of global and local models by introducing regularization terms into local optimization objectives. pFedMe (T Dinh et al., 2020) used the Moreau envelope (Moreau, 1965) as a client-side regularization term, separating the optimization of personalized models from global model learning, theoretically achieving faster convergence rates. Moreover, to tackle parameter inconsistency in personalized FL, some studies adopt a consistency alignment optimization strategy. Li et al. (2024c) incorporated local batch normalization layers into client models to adapt to local data distributions without altering the global model structure, thus enhancing model personalization performance. The FedAS (Yang et al., 2024b) framework used the trace of the Fisher information matrix to measure client training progress and performs weighted aggregation accordingly, effectively reducing the negative impact

of clients with slower training progress on the global model. More lightweight methods (Tan et al., 2022; Yao et al., 2026) rely on communicating global representation prototypes across clients to aggregate shared knowledge rather than using feature extractors.

**Client Grouping.** Client grouping is an effective strategy for addressing client heterogeneity in TPFL. Early studies, such as FedCS (Han et al., 2023), introduced a confidence-based, similarity-aware strategy that enables flexible client grouping and similarity weight adjustment. Subsequent research has explored more advanced clustering techniques. MiniPFL (Fan et al., 2024) used the BIRCH clustering algorithm to effectively distinguish similar clients and conduct collaborative training within divided "mini federations", enhancing the efficiency of server-side federated updates. Other studies leverage endogenous structures within systems or networks for explicit clustering. CA-PFL (Zhao et al., 2024b) constructed a graph-structured federated social network and employs community detection to group clients. This ensures that clients within the same community have similar label representations, allowing for the aggregation of shared layers within each community while retaining personalized layers for each client to address concept drift. Moreover, client screening can be seen as a special binary clustering method, aiming to create a homogeneous group of clients. Ge et al. (2024) proposed a one-shot screening scheme that converts local losses of clients on a pre-trained model into p-values and uses adaptive thresholds to filter out client groups with homogeneous data distributions to the task initiator, thereby tailoring FL for specific tasks.

### 4.1.2 Evaluation

**Datasets.** The commonly used experimental datasets for TPFL, as shown in Table 3 include Tiny ImageNet, Omniglot, Shakespeare, CelebA, and Sent140.

**Setup.** Under heterogeneous task scenarios, clients are assumed to be divided in different proportions to perform various types of tasks. Most existing studies tested five types of learning tasks: object detection, semantic segmentation, image classification, object tracking, and natural language processing related tasks. Clients are randomly assigned to these tasks according to preset proportions. Meanwhile, to simulate the real-world data heterogeneity, even clients performing the same type of task can be assigned different datasets. In addition, each dataset is distributed to clients in a non-IID manner: for label types, skewed sampling is performed using the Dirichlet distribution; regarding data volume, there is a significant imbalance among clients, and datasets of different sizes are split and allocated to each client. Each client may adopt a distinct model architecture tailored to its task type and dataset characteristics, while the aggregability among models is ensured through a unified feature representation layer.

**Metric.** Two metrics are usually used: average accuracy and average task loss of all clients' local test datasets. Besides, specific metrics are used for specific tasks.

### 4.1.3 Real-world Applications

Several representative applications highlight the practical value of TPFL across various domains. In healthcare, ProtoHAR (Cheng et al., 2023) developed a prototype-guided mechanism for privacy-preserving and individualized human activity recognition in wearable healthcare systems, while PPFL (Kim et al., 2024) proposed a personalized progressive FL approach to leverage institution-specific features for collaborative medical diagnosis across different healthcare institutions. An adaptive FL approach to break data silos for Parkinson's disease diagnosis via privacy-preserving facial expression analysis was proposed in (Pang et al., 2025b). A tri-factor adaptive FL framework for Parkinson's disease diagnosis via multi-source facial expression analysis was introduced in (Pang et al., 2025a). In edge computing scenarios, Ge et al. (2024) proposed a one-shot screening mechanism to identify data-homogeneous edge users for personalized FL in edge intelligent networks. In industrial applications, FedScrap (Zhang et al., 2024e) adopted a layer-wise personalization strategy for customized model training in industrial waste detection scenarios.

## 4.2 Self-Supervised Federated Learning

Self-Supervised FL (SSFL) stands out as another crucial approach in improving the cross-task generalization ability of FL models, especially in environments where labeled data is scarce or unavailable. SSFL is

not dedicated to certain explicit tasks; instead, it aims to learn a general global model that can extract task-generalizable representations from certain input features. Specifically, each client $\mathcal{C}_k$ conducts the same self-supervised learning (SSL), such as contrastive learning and rotation prediction, on its local data $\mathcal{D}_k$ to optimize the local model $m'_k$ with the loss $\mathcal{L}_{\text{self}}\left[m'_k(\mathcal{X}_k)\right]$. Overall, SSFL enables the extraction and integration of diverse, self-generated features and knowledge across clients, which is beneficial for incorporating new tasks into FL systems. To facilitate a clear comparison, we summarize representative SSFL approaches in Table 7, evaluating their support for different learning scenarios (timing, periodicity, and synchronism), associated overheads, and overall utility.

Table 7: Comparison of approaches in Self-Supervised Federated Learning. While **data reconstruction** methods typically incur high computational and communication costs, **contrastive learning** approaches effectively address these resource constraints by offering superior system efficiency and reduced overheads.

| Category | Methods | Timing Support | | Period Support | | Synchronism | | Overhead | | | Utility |
|---|---|---|---|---|---|---|---|---|---|---|---|
| | | Abrupt | Graduate | Short | Long | Sync | Async | Computation | Communication | Storage | Performance |
| Contrastive Learning | MocoSFL (Li et al., 2022b) | ✓ | ✗ | ✗ | ✓ | ✓ | ✗ | Medium | High | High | Behind SOTA |
| | FedSC (Jing et al., 2024) | ✓ | ✗ | ✗ | ✓ | ✓ | ✗ | Medium | Medium | Low | Near SOTA |
| | AeroRec (Xia et al., 2024a) | ✓ | ✓ | ✓ | ✓ | ✓ | ✗ | Low | Low | Low | SOTA |
| | FedCAD (Kong et al., 2024) | ✓ | ✗ | ✗ | ✓ | ✓ | ✗ | High | Medium | Low | SOTA |
| | FLSimCo (Gu et al., 2024) | ✓ | ✓ | ✓ | ✓ | ✓ | ✗ | Low | Medium | Low | SOTA |
| Data Reconstruction | FedIICWu et al. (2022) | ✓ | ✓ | ✗ | ✓ | ✓ | ✗ | High | High | Medium | Behind SOTA |
| | FedMAE (Yang et al., 2023b) | ✓ | ✓ | ✗ | ✓ | ✓ | ✗ | High | High | Medium | Near SOTA |
| | FedNI (Peng et al., 2022) | ✓ | ✗ | ✗ | ✓ | ✓ | ✗ | High | Medium | Medium | Near SOTA |
| | GraphFL (Wang et al., 2022a) | ✓ | ✗ | ✗ | ✓ | ✓ | ✗ | Medium | Medium | Low | Behind SOTA |
| | MS-DINO (Park et al., 2024) | ✓ | ✓ | ✗ | ✓ | ✓ | ✗ | High | High | High | SOTA |
| | FedRecon (Liu et al., 2025b) | ✓ | ✓ | ✗ | ✓ | ✓ | ✗ | High | High | High | SOTA |

### 4.2.1 Methods

**Contrastive Learning.** Contrastive learning aims to learn discriminative feature representations by maximizing the consistency between similar samples while minimizing the similarity among dissimilar ones. Early studies primarily focused on effective sample comparisons. MocoSFL (Li et al., 2022b), based on the momentum contrast mechanism, maintains a shared feature memory and employs frequent synchronization strategies, laying the groundwork for contrastive methods in SSFL. In parallel, by sharing correlation matrices of data representations across clients, FedSC (Jing et al., 2024) enabled cross-client sample comparisons. To further improve efficiency, AeroRec (Xia et al., 2024a) combined contrastive learning with knowledge distillation techniques, proposing a contrastive distillation approach for efficient knowledge transfer. Moreover, significant progress has also been made in contrastive learning tailored to specific data types. FedCAD (Kong et al., 2024) extended contrastive SSL to graph data by aggregating embeddings of neighboring nodes around anomalous nodes, enhancing the discriminability between positive and negative sample pairs, and providing a novel solution for federated graph learning. FLSimCo (Gu et al., 2024) considered the impact of data quality on contrastive learning, proposing a dynamic aggregation strategy based on image blur levels, demonstrating the adaptability of contrastive learning with low-quality data.

**Data Reconstruction.** Apart from contrastive learning, other approaches typically use data reconstruction for SSL. Wu et al. (2022) introduced the masked autoencoder reconstruction task into FL, achieving effective feature learning through the auxiliary task of reconstructing masked image regions. Yang et al. (2023b) proposed a more general FedMAE framework, extending this pixel reconstruction strategy from specific medical scenarios to general visual tasks. However, pixel reconstruction mainly focuses on low-level texture information and has limited ability to model high-level relationships in structured data. FedNI (Peng et al., 2022) extended the idea of reconstruction to graph data, capturing more complex topological structural relationships by predicting missing node and edge information through network repair tasks. GraphFL (Wang et al., 2022a) provided an FL framework for semi-supervised node classification on graph-structured data. With the widespread application of transformer architecture, Park et al. (2024) proposed MS-DINO that performs distributed SSL by predicting masked image patches. In multi-modal scenarios, Liu et al. (2025b) proposed the FedRecon framework, which performs SSL by reconstructing missing modal information.

### 4.2.2 Evaluation

**Datasets.** The commonly used datasets, as shown in Table 3 include Tiny ImageNet, Shakespeare, CelebA, Sent140, SVHN, COVID-FL, Kinetics 400, and UCF101.

**Setup.** First, a federated environment is simulated by dividing the dataset among a group of clients. This partitioning typically uses the Dirichlet distribution to manage data heterogeneity among clients. The experiment is divided into two stages: the first is the collaborative self-supervised pre-training stage. During this stage, all clients use only their local unlabeled data to jointly optimize a shared feature extractor. In each communication round, clients perform SSL tasks locally, then send the updated model parameters to the server for aggregation. After pre-training, the downstream task evaluation stage begins. At this point, the shared feature extractor is frozen, and the quality of the features is evaluated using a reserved, small-scale labeled dataset (e.g., 1% or 10% of the total training data).

**Metric.** Two metrics are used to evaluate the performance of SSFL: linear probe accuracy is the accuracy of using a small number of labeled data to fine-tune a linear classifier based on the SSL-pretrained feature extractor; K-nearest neighbor classification accuracy is the accuracy of assuming the classifier is a KNN.

### 4.2.3 Real-world Applications

In transportation scenarios, Dai et al. (2024) proposed using prototype clustering contrastive learning for traffic scene recognition and vehicle behavior analysis, while Soares et al. (2025) proposed an SSFL approach with Bayesian optimization for monocular depth estimation in autonomous vehicle applications. In medical diagnosis, Sun et al. (2025) proposed a hierarchical semi-supervised FL method for dermatosis diagnosis. In industrial monitoring, Youn et al. (2025) proposed a self-supervised asynchronous FL approach for diagnosing partial discharge in gas-insulated switchgear for power system monitoring.

## 4.3 Future Work

Future research on integrating new tasks into FL can focus on constructing a more intelligent, efficient, and robust learning system. An ideal FL framework should possess the full-chain autonomous adjustment capability from task characteristic perception, cross-task knowledge transfer to model adaptive optimization when facing dynamically emerging new tasks. It should not only maintain generalization performance in scenarios where heterogeneous data and complex tasks are interwoven, but also adapt to the resource constraints of edge devices through lightweight design, while taking into account privacy security and system scalability under multi-task collaboration. The specific discussions are carried out from the two major directions of TPFL and SSFL as follows:

**TPFL.** Future TPFL research should aim to overcome current limitations in handling dual heterogeneity of tasks and data among clients, and develop new theoretical models and algorithm frameworks. First, within knowledge sharing and personalization paradigms, although existing works balance sharing and personalization using adaptive masks (Lv et al., 2024) or hypernetworks (Scott et al., 2024), a revolutionary direction is to develop a federated multi-task learning mechanism based on causal inference. This can achieve precise knowledge transfer by identifying causal relationships between tasks, surpassing traditional correlation analysis. Second, regarding system architecture and collaboration mechanisms, decentralization is crucial for enhancing scalability (Feng et al., 2025a). By evolving TPFL system architecture, self-organizing and self-healing federated network topologies can be designed, allowing clients to form optimal collaboration patterns via smart contracts without central coordination. Finally, to resolve inter-task interference, current methods focus on suppressing interference through aggregation strategies (Wei et al., 2025). A viable solution could be constructing model representations by encoding knowledge of different tasks into orthogonal neural subspaces, ensuring minimal impact on old tasks and enabling forgetfulness-free lifelong task learning.

**SSFL.** One potential future direction is to develop a multimodal SSFL framework to address differences in modal combinations among clients. Initial progress includes CroSSL (Deldari et al., 2024), which introduced cross-modal SSL via latent masking for time series, handling motion sensors and biological signals without negative pair sampling. Zhang et al. (2024g) used complementary data and pseudo-labeling algorithms to improve cross-modal learning. Secondly, the high computational and communication costs incurred by

SSFL algorithms are issues that also need to be addressed. For instance, LW-FedSSL (Tun et al., 2024) adopted a layer-wise incremental training strategy, allowing edge devices to train one model layer at a time. Combined with server-side calibration and representation alignment, it significantly reduces SSFL algorithms' computational and communication overhead. Additionally, the problem of catastrophic forgetting is more complex in the SSFL environment, so future systems need strong continuous learning capabilities.

## 5 Incorporate New Tasks with New Features

In practical applications of FL, the arrival of new tasks is often accompanied by new features. Typically, learning new tasks based on the models that have already undergone some training is referred to as continual learning (CL). As a result, it is necessary to extend CL to the FL context, achieving Federated Continual Learning (FCL) (Yang et al., 2023d). While regular CL is not limited to classification, existing research in FCL nearly all assumes that new tasks are presented as new data classes. Therefore, we use classification as a representative example to elucidate FCL here. We make a systematic review of current technical approaches in FCL and compare them in the supported arrival timing and existing period of new knowledge, overhead, and performance, shown in Table 8.

Table 8: Comparison of approaches in Federated Continual Learning. All these methods cannot support the effective learning when the new knowledge data exists for a short period. Considering the overhead and performance, **Architecture Decomposition** presents as a promising solution.

| Category | Methods | Timing Support | | Period Support | | Synchronism | | Overhead | | | Utility |
|---|---|---|---|---|---|---|---|---|---|---|---|
| | | Abrupt | Graduate | Short | Long | Sync | Async | Computation | Communication | Storage | Performance |
| Optimization Regularization | FedCurv (Shoham et al., 2019b) | ✓ | × | × | ✓ | ✓ | × | Medium | High | High | Behind SOTA |
| | CPPFL (Huang et al., 2022b) | ✓ | × | × | ✓ | ✓ | × | Medium | Medium | Low | Behind SOTA |
| | FedCL (Yao & Sun, 2020) | ✓ | × | × | ✓ | ✓ | × | High | High | High | Near SOTA |
| | FedSyR (Li et al., 2024e) | ✓ | × | × | ✓ | ✓ | × | High | Medium | Medium | Near SOTA |
| | FLwF (Usmanova et al., 2021) | ✓ | × | × | ✓ | ✓ | × | High | High | High | Behind SOTA |
| | GLFC (Dong et al., 2022) | ✓ | ✓ | × | ✓ | ✓ | × | High | High | High | Near SOTA |
| | FCCL (Huang et al., 2022a) | ✓ | × | × | ✓ | ✓ | × | High | High | Medium | Near SOTA |
| | FedC3DS-SR(Peng et al., 2025) | ✓ | × | × | ✓ | ✓ | × | High | High | Medium | SOTA |
| | CGoFed (Feng et al., 2025b) | ✓ | × | × | ✓ | ✓ | × | High | High | Low | SOTA |
| Knowledge Replay | Re-Fed+ (Li et al., 2025b) | ✓ | × | × | ✓ | ✓ | × | High | High | High | Behind SOTA |
| | DP-FCL-DS (Zizzo et al., 2022) | ✓ | ✓ | × | ✓ | ✓ | ✓ | High | High | Medium | Behind SOTA |
| | FCIL (Sun et al., 2024) | ✓ | ✓ | × | ✓ | ✓ | ✓ | High | High | Low | Near SOTA |
| | FedGR (Qi et al., 2022) | ✓ | × | × | ✓ | ✓ | × | High | High | Medium | Near SOTA |
| | FedMAE-CL (He & Wang, 2024) | ✓ | × | × | ✓ | ✓ | × | High | High | Medium | Near SOTA |
| | FedDDR (Liang et al., 2024) | ✓ | × | × | ✓ | ✓ | × | High | High | Medium | SOTA |
| | FedDGR (Mei et al., 2024) | ✓ | × | × | ✓ | ✓ | × | High | High | Medium | SOTA |
| | FedGTG (Nguyen et al., 2024b) | ✓ | ✓ | × | ✓ | ✓ | × | High | High | Low | SOTA |
| | AF-HFCL (Wuerkaixi et al., 2025) | ✓ | ✓ | × | ✓ | ✓ | × | High | High | Low | SOTA |
| | NGFL (Li et al., 2022a) | ✓ | ✓ | × | ✓ | ✓ | ✓ | High | High | Medium | Behind SOTA |
| | PF-FCIL (Yoo & Park, 2024) | ✓ | ✓ | × | ✓ | ✓ | ✓ | High | High | Medium | Near SOTA |
| | FedMR (Wu et al., 2025b) | ✓ | ✓ | × | ✓ | ✓ | ✓ | High | High | High | SOTA |
| | AFL-IIoT (Zhang et al., 2025a) | ✓ | ✓ | × | ✓ | ✓ | ✓ | High | High | High | SOTA |
| Architecture Decomposition | FedSeIT (Chaudhary et al., 2022) | ✓ | × | × | ✓ | ✓ | × | High | Medium | High | Behind SOTA |
| | Loci (Luopan et al., 2025) | ✓ | × | × | ✓ | ✓ | × | High | Medium | Medium | SOTA |
| | FedTA (Yu et al., 2024) | ✓ | × | × | ✓ | ✓ | × | High | Medium | Medium | Near SOTA |
| | FOT (Bakman et al., 2024) | ✓ | × | × | ✓ | ✓ | × | High | Low | Medium | Near SOTA |
| | FedPDT (Piao et al., 2024) | ✓ | × | × | ✓ | ✓ | × | High | Medium | Medium | SOTA |

### 5.1 Definition of Federated Continual Learning

#### 5.1.1 Synchronous FCL

Similar to standard CL, Synchronous FCL assumes that there are a series of datasets $\{\mathcal{D}^t\}_{t=1}^T$ with $N^t$ data samples $\mathcal{D}^t = \{(\boldsymbol{x}_j^t, y_j^t)\}_{j=1}^{N^t}$ continuously arriving in the FL systems globally. The majority of each dataset is assumed to belong to previously unseen data classes, i.e., $\mathcal{Y}^t \cap \cup_{i=1}^{t-1}\mathcal{Y}^i = \emptyset$. Moreover, actually, the dataset at each time $t$ is also non-IID owned by a part of clients at that time, i.e., each client $\mathcal{C}_k$

possesses $\mathcal{D}_k^t = \{(\boldsymbol{x}_{k,j}^t, y_{k,j}^t)\}_{j=1}^{N_k^t}$ and the corresponding marginal distributions are distinct among clients, $\mathcal{X}_k^t \neq \mathcal{X}_{k'}^t, \mathcal{Y}_k^t \neq \mathcal{Y}_{k'}^t$ (where $k \neq k'$, but $\mathcal{Y}_k^t, \mathcal{Y}_{k'}^t \subseteq \mathcal{Y}^t$). Consistent with existing Synchronous FCL studies, we also consider practical constraints from limited storage space and privacy regulations (Wang et al., 2022b; Guo et al., 2023), and assume that old class data is unavailable or can be only accessed partially by the clients that own new class data ($\mathcal{C}_{\mathrm{new}}$ and $\mathcal{C}_{\mathrm{mix}}$). Then the objective is to minimize the classification errors for new classes while preserving the good performance of learned classes

$$\min_{\theta^t, \omega^t} \mathbb{E}_{\boldsymbol{x} \sim \cup_{i=1}^{t-1} \mathcal{X}^i} \left[ \|g_{\omega^t}(f_{\theta^t}(\boldsymbol{x})) - g_{\omega^{t-1}}(f_{\theta^{t-1}}(\boldsymbol{x}))\|^2 \right] + \mathbb{E}_{(\boldsymbol{x}, y) \sim (\mathcal{X}^t, \mathcal{Y}^t)} \left[ \mathcal{L}(g_{\omega^t}(f_{\theta^t}(\boldsymbol{x})), y) \right]. \tag{1}$$

### 5.1.2 Asynchronous FCL

Different from Synchronous FCL, Asynchronous FCL assumes that FL clients learn a series of classes in their own distinct orders. Specifically, suppose there is a set of class spaces $\{\mathcal{Y}^t\}_{t=1}^T$ with the order $\mathcal{O} = [1, ..., t, ..., T]$ from the global perspective, and these class spaces are disjoint from each other $\mathcal{Y}^t \cap \mathcal{Y}^{t'} = \emptyset$ (where $t \neq t'$). However, from the local perspective, each client $\mathcal{C}_k$ continually receives the dataset corresponding to a particular order $\mathcal{O}_k = [1_k, ..., t_k, ..., T_k]$. The orders of clients are distinct from each other as well as from the global one, i.e., $\mathcal{O}_k \neq \mathcal{O}_{k'}, \mathcal{O}_k \neq \mathcal{O}$ (where $k \neq k'$). Asynchronous FCL also conforms to the constraints of limited storage space and privacy regulations. As for the difference between different client categories, we assume that $\mathcal{C}_{\mathrm{old}}$'s data is fixed until the category changes into others, while the data of both $\mathcal{C}_{\mathrm{new}}$ and $\mathcal{C}_{\mathrm{mix}}$ is changing over time as long as they do not change into $\mathcal{C}_{\mathrm{old}}$. Then, the objective of Asynchronous FCL is to minimize the classification errors for new classes without or with partial access to previously learned classes on the local side, which can be viewed as a local personalization of Eq. (1).

### 5.2 Methods

**Optimization Regularization.** During CL, optimizing model parameters for new tasks can lead to forgetting previously learned tasks. Adding regularization to limit large parameter changes helps alleviate this forgetting. Fisher information and synaptic intelligence are commonly used to assess the importance of model parameters. FedCurv (Shoham et al., 2019b) and CPPFL (Huang et al., 2022b) were pioneers in applying these methods to local FL training. However, they do not consider the impact of data heterogeneity in FL, which is the main focus of Yao & Sun (2020). Li et al. (2024e) also addressed this issue by modifying synaptic intelligence to reflect model parameter importance based on local datasets and their correlation to the global data distribution. Model parameter change is also reflected as intermediate layer output change, thus, another forgetting alleviation is to conduct knowledge distillation for preserving learned task knowledge. FLwF (Usmanova et al., 2021) first built a teacher-student framework among a stored old task model and the current model, and regularizes similar logits over the same input, which corresponds to the first term of Eq. (1). GLFC (Dong et al., 2022) designed a weighting mechanism for logit regularization to alleviate the impact of data heterogeneity. In contrast, FCCL (Huang et al., 2022a) assumed a public dataset could be used to distill knowledge from global old models, a method also adopted by Peng et al. (2025). Model parameter updates align with model gradients during training. Thus, CGoFed (Feng et al., 2025b) proposed constraining the gradient update direction to a space that minimizes interference with historical tasks, helping reduce forgetting of old data and speeding up adaptation to new data.

**Knowledge Replay.** The forgetting of learned tasks occurs because the model is optimized only for new tasks. This raises the question: 'Can we incorporate objectives for learned tasks into the optimization process?' Accordingly, replay-based solutions are proposed, which replay old tasks during FCL. Intuitively, random storing can be inefficient; thus, GLFC (Dong et al., 2022) tried to store the samples that are closest to each class center. However, Re-Fed+ (Li et al., 2025b) argued that data heterogeneity should be considered when selecting the data to store, and proposed to use a personalized informative model to ensure the selected data contributes to both local and global knowledge. In addition to such local storage of old class data, Zizzo et al. (2022) built a global memory bank hosted on the central server to better deal with the global forgetting caused by data heterogeneity. However, managing this global memory requires clients to upload their private data, which violates the privacy preservation of FL, Zizzo et al. (2022) introduced Laplace noise to protect uploaded data. Instead of uploading the raw data, Sun et al. (2024) leveraged dataset distillation to condense the local old task data by ensuring gradient consistency between the original and condensed data. Other

works enabled training generative models like GAN (Qi et al., 2022), masked autoencoders (He & Wang, 2024), and diffusion models (Liang et al., 2024; Mei et al., 2024) locally to generate old class data for replay. However, communicating these locally trained generative models also results in privacy leakage; thus, Nguyen et al. (2024b) used data reconstruction attacks for good to train a global old data generator on the server side at the end of each task training, which generates data to be replayed in future task training. Besides, different from data sample replay, Wuerkaixi et al. (2025) used a normalization flow model to generate old task features for replay while learning new tasks. Similar ideas can also be found in Li et al. (2022a); Yoo & Park (2024), which replayed old class prototypes across clients and the server. Besides, not only can the data be replayed, Wu et al. (2025b) proposed replaying historical client models and developed a weighted distillation mechanism to alleviate forgetting of old tasks. This model-based replay strategy was also adopted by Zhang et al. (2025a).

**Architecture Decomposition.** To preserve the old task knowledge, one intuitive approach is to separate different tasks and assign them to distinct model parameters. FedSeIT (Chaudhary et al., 2022), a natural language processing study, divided the model parameters into separate parts. Except for similar task-general and specific parts, there is another part to measure the correlation between tasks in FedSeIT, thus it can spare the parameter bank. Similar ideas are also adopted by Lu et al. (2024) and Wu et al. (2025a). In fact, the model decomposition introduces additional communication cost. To reduce the cost, Loci (Luopan et al., 2025) proposed to compress the task-shared models into compact versions while leaving task-specific models untransmitted. From the perspective of feature sensitivity, FedTA (Yu et al., 2024) was built on the observation that the last few model layers are associated with task variations, and achieved FCL by adjusting the locations of task-shared and task-specific representations in the latent space. FOT (Bakman et al., 2024) modified the new task training to make its model updates orthogonal to the previous task activation principal subspace, which can prevent interference between tasks. Although this strict model separation prevents interference, it also discards the inter-task performance enhancements due to their positive knowledge correlations. Piao et al. (2024) proposed to separate the positive and negative knowledge correlations between tasks in model parameters and then amplify the positive transfer while mitigating the negative transfer.

### 5.3 Evaluation

**Datasets.** The commonly used experimental datasets for FCL, as shown in Table 3, include FEMNIST, CIFAR-10/CIFAR-100, Tiny ImageNet, Omniglot, and CelebA.

**Setup.** In a typical FCL setup, a dataset is first partitioned into a predefined number of sequential tasks based on data categories. A common strategy is uniform sequential allocation, where each task is assigned an equal number of non-overlapping classes, following the dataset's original class order. More flexible setups relax these constraints, for instance, by randomly determining the number of classes per task and which specific classes are assigned. For any given task, the data is then distributed across clients in a non-IID manner. This is often achieved by inducing label skew, where each client only observes data for a subset of the task's classes. A widely adopted method to model this is to use a Dirichlet distribution to allocate class proportions to each client. The experimental setup also defines the task arrival sequence and duration. In synchronous FCL, all clients encounter the same sequence of tasks. In contrast, asynchronous FCL allows each client to learn from tasks in a different, often randomized, order. As for task duration, most studies assume a fixed period for each task (e.g., a set number of epochs), thus obviating the need for task-boundary detection. However, a growing body of work is focused on designing sensitive, real-time detectors to enable FCL to operate under more realistic, arbitrary task boundaries.

**Metrics.** Two metrics are used to measure the performance of FCL: average accuracy (ACC) and forgetting (FGT) (Bakman et al., 2024),

$$\text{ACC} = \frac{1}{t} \sum_{i=1}^{t} a^{i,t}, \quad \text{FGT} = \frac{1}{t-1} \sum_{i=1}^{t-1} a^{i,i} - a^{i,t}, \tag{2}$$

where $a^{i,t}$ is the model accuracy of task $\mathcal{Y}^i$ right after learning task $\mathcal{Y}^t$. As for Asynchronous FCL, ACC, and FGT of local clients are also used to measure the performance.

### 5.4 Real-world Applications

Some preliminary studies may provide valuable insights. FCLLM-DT (Xia et al., 2024b) built a digital twin physical model to synthesize virtual anomaly data, which is integrated with FCL to realize spatial and temporal anomaly detection in the industrial Internet of Things. Similar tries can be seen in the intrusion detection (Zhang et al., 2024h; Quyen et al., 2024; Mao et al., 2024) for industrial manufacturing systems. In multi-satellite networks, FCIL-MSN (Niu et al., 2024) achieved FCL in satellite scene recognition.

### 5.5 Future Works

First of all, potential future studies involve extending FCL to multi-label and multi-grained classification. Besides, some preliminary works have explored achieving federated incremental segmentation on medical data (Peng et al., 2025; Zhang et al., 2024a; Dong et al., 2026). Therefore, extension to more complicated tasks like object tracking and visual question answering is waiting to be explored. Additionally, considering FCL in practical scenarios is underexplored, such as learning new tasks with insufficient, weakly labeled, or unlabeled data. Although there are a few studies related to few-shot FCL (Liang et al., 2025a) and federated novel class discovery, their effectiveness usually relies on unrealistic assumptions. In addition, developing FCL techniques for more data modalities is needed. Fortunately, some examples in graph classification (Miao et al., 2025; Zhu et al., 2024) and text entity recognition (Zhang et al., 2025c) have been witnessed. Employing FCL to tackle some fundamental challenges of FL is another potential direction. Wang et al. (2024a) constructed a digital twin model for each FL client to generate data continuously, addressing non-IID issues in FL. Future research could explore leveraging FCL to reduce computation costs and accelerate model convergence. Besides, more applications of FCL in healthcare, finance, and even scientific discovery are worthy of research.

## 6 Incorporate New Models

Integrating a broader range of advanced models, including Foundation Models (FMs) and Large Language Models (LLMs), into existing FL systems significantly enhances traditional FL frameworks (Zhuang et al., 2023). Traditional FL systems often struggle with complex data distributions and the demands of sophisticated tasks. Incorporating models with superior architecture and larger scales is crucial. This approach not only addresses the limitations of older FL systems but also promotes evolution by increasing data availability, boosting collaborative development, and enhancing both the utility and privacy of FL models in various applications. In the following context, we consider two cases for incorporating new models $\mathcal{M}_{\text{new}}$ into FL: (i) **From New to Old:** use new models $f_\alpha$ (e.g., FMs and LLMs) to enhance existing models $f_\theta$; (ii) **From Old to New:** transfer architecture and knowledge (e.g., from convolution neural networks to transformers) from the existing models $f_\theta$ to new models $f_\alpha$. In both cases, the new model $f_\alpha$ is assumed to be pre-trained on certain datasets outside the target FL systems. In the first scenario, clients can use their local data to distill knowledge from $f_\alpha$, which is typically hosted on the central server $\mathcal{S}$ (as $f_\alpha$ is usually quite large). In the second scenario, the focus is on efficiently transferring old knowledge from $f_\theta$ to provide a better starting point for optimizing $f_\alpha$. Beyond these two transfer directions, new model architectures can also act as a flexible interface to *integrate continuously arriving knowledge* (e.g., new features or new tasks) into the evolving FL system. Accordingly, we organize the technical approaches in this section into two complementary objectives: (i) enhancing current functionality via model transfer (Section 6.1.1); and (ii) facilitating new knowledge integration via architectural modularity and adaptation mechanisms (Section 6.1.2). The potential technical approaches are reviewed below, shown in Figure 3.

### 6.1 Methods

#### 6.1.1 For Enhancing Current Functionality

**From New to Old.** When transferring models from new to old, two major issues need consideration: 1) local client data alone is insufficient to fully utilize the knowledge in new models; 2) new models are typically too large to be loaded on clients or communicated efficiently between clients and the server. For the first issue, CROSSLM (Deng et al., 2023) demonstrated using small language models to refine LLMs

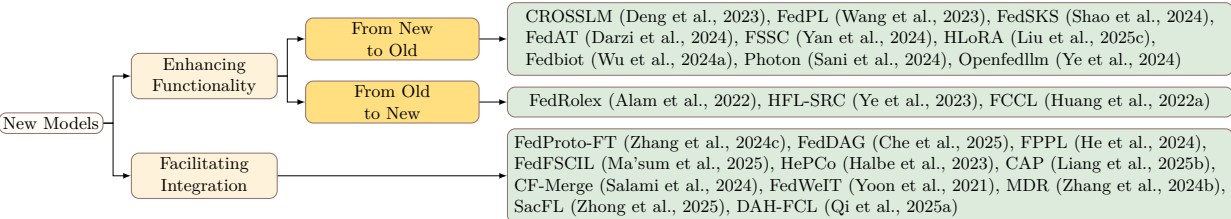

Figure 3: A taxonomy of technical approaches for incorporating new models into FL systems can be organized into two broad categories: (i) methods that enhance existing functionalities, and (ii) methods that facilitate the integration of additional new knowledge. For the former, we consider architectural transfer either from the new model to the old model or from the old model to the new model.

and generate synthetic data, enabling data-free knowledge transfer. Additionally, approaches such as using public data and LLMs to enhance federated distillation by accurately selecting knowledge from local and ensemble predictions, as investigated by Wang et al. (2023) and Shao et al. (2024), show the potential for knowledge transfer from new models to established FL models. Beyond distilling knowledge from the model's output layer, another approach focuses on aligning intermediate feature representations. Darzi et al. (2024) introduced a feature alignment loss to ensure that the intermediate representations of local models align with a global vision transformer (ViT), addressing unfair enhancements for clients with limited data. For the second issue, one solution is adopting more communication-efficient model architectures. For instance, Yan et al. (2024) integrated small-scale Swin Transformers into FL instead of regular ViT. A more common approach involves parameter-efficient fine-tuning techniques, which avoid updating and communicating the entire model. Liu et al. (2025c) introduced low-rank adaptation into FL, allowing clients to learn and upload only a small adaptation matrix, significantly reducing communication and computation overhead. Similarly, Wu et al. (2024a) designed a mechanism for on-demand parameter transmission, where only necessary updates are exchanged, achieving performance close to centralized fine-tuning with minimal communication cost. In addition to selective parameter updates, other works have integrated dedicated optimization techniques to address communication bottlenecks. For example, the Photon framework (Sani et al., 2024) combined gradient quantization, dynamic communication frequency adjustment, and efficient ring-based parameter synchronization to reduce bandwidth needs significantly. Similarly, Ye et al. (2024) developed an open-source FL framework for LLMs that optimizes communication compression and parallel scheduling, further addressing the challenges of training large models over low-bandwidth networks.

**From Old to New.** When transferring from old models to new ones, existing FL studies on heterogeneous model architectures can be beneficial. For example, FedRolex (Alam et al., 2022) implemented a rolling window mechanism that allows clients to selectively extract sub-models from the global model on the central server, tailoring them to train different local models. Similar concepts were explored in the work of Ye et al. (2023). Another approach (Huang et al., 2022a) used a public dataset and SSL to distill knowledge from old FL models to new ones with heterogeneous architectures. However, this requires the public dataset to closely match the learned data distribution, which is often impractical. In future research, data availability will remain a significant challenge. Not all learned data can be stored for transferring to new architectures, necessitating selective data storage. Additionally, if the new architectures are large, specific designs for split learning or tuning are needed, such as using adapters for different attention layers when training transformers.

### 6.1.2 For Facilitating New Knowledge Integration

Beyond enhancing existing capabilities, new model architectures increasingly serve as a flexible interface to facilitate the integration of other types of new knowledge, effectively transforming the global model into a universal foundation. To facilitate the integration of new features, particularly unseen distributions, architectures are evolving to include auxiliary alignment components. For instance, Zhang et al. (2024c) employed global adapters trained on multi-domain prototypes to structurally bridge heterogeneous feature spaces, while Che et al. (2025) incorporated domain-simulation modules to actively synthesize features for alignment. Conversely, when incorporating new tasks, structural modularity becomes essential to manage

sequential knowledge without forgetting. Prompt-based architectures offer a lightweight solution: He et al. (2024), Ma'sum et al. (2025), and Wu et al. (2025c) utilized task-specific prompts to manage sequential new tasks, while Halbe et al. (2023) and Liang et al. (2025b) leveraged prompt pools to enable efficient task adaptation. Similarly, Salami et al. (2024) trained low-rank adapters to extend model capacity for specific tasks. Decomposing the architecture into decoupled components is another robust pattern. FedWeIT (Yoon et al., 2021) and pFedC (Zhang et al., 2024b) demonstrated that by separating parameters into "task-general" and "task-specific" categories or using conflict-aware masks, the system can selectively integrate new knowledge. Furthermore, SacFL (Zhong et al., 2025) adopted an encoder-decoder split to facilitate continuous task arrival. Finally, Qi et al. (2025a) utilized dynamic hypernetworks to learn the correspondence between task identities and model weights, automating the architectural adaptation to continuously arriving tasks.

## 6.2 Real-world Applications

On the one hand, transitioning from new to old models, Galal et al. (2024) proposed using pre-trained BERT to enhance existing FL models in distributed text classification. In drug discovery, Wang et al. (2025b) introduced a federated knowledge distillation approach to improve the accuracy of compound-protein interaction predictions through collaborative learning across pharmaceutical institutions. For time series analysis, Chen et al. (2025b) suggested leveraging foundation models to enhance FL approaches for heterogeneous time series data across distributed clients. On the other hand, transitioning from old to new models, in edge computing scenarios, Zuo et al. (2024) proposed a federated continual learning approach to transfer knowledge from traditional CNN models to ViT for improved visual recognition on edge devices. In medical imaging, Tölle et al. (2025) proposed transferring knowledge from existing CNN-based models to advanced transformer architectures for enhanced cardiac CT image analysis in FL settings.

## 6.3 Future Works

Current federated knowledge distillation methods face significant communication and computational challenges, and they lack robustness in diverse environments. These methods primarily depend on transmitting model outputs (logits) for distillation. Future research should investigate more efficient knowledge representation forms. For instance, studies could explore how to distill and transmit compressed or quantized knowledge (He et al., 2025b), or design protocols that only transmit the most essential knowledge (Gad et al., 2024). Additionally, when enhancing existing models with the general knowledge of foundation models, there is a risk of overwriting valuable expertise acquired in specific tasks. Future work should incorporate continuous learning into knowledge distillation. For example, using techniques like elastic weight consolidation (Shoham et al., 2019a) or knowledge replay (Pennisi et al., 2024), new knowledge can be introduced while effectively preserving and balancing existing expertise. Model evolution expands the capacity and interface of an FL system, but system-level evolution also depends on the coordination logic that turns heterogeneous client updates into global progress. As client populations, data heterogeneity, and privacy constraints change over time, an established FL system may need to upgrade its aggregation rule to preserve existing functionality while absorbing newly arriving knowledge. We therefore next review aggregation algorithms as a fourth carrier of new knowledge in evolving FL systems.

## 7 Incorporate New Aggregation Algorithms

Aggregation algorithms define the coordination logic of FL systems: they determine how client updates are weighted, synchronized, protected, and ultimately translated into global model improvements. As FL deployments evolve, naive averaging becomes increasingly insufficient under practical constraints such as severe non-IID data, system heterogeneity, privacy and security requirements, and continuously changing client populations. Consequently, incorporating new aggregation algorithms is not merely a performance tweak but a key mechanism for enabling system-level evolution. In this section, we review aggregation methods from two complementary perspectives: (i) **For Enhancing Current Functionality,** where new algorithms improve robustness, convergence, and privacy guarantees under existing learning objectives; and (ii) **For Facilitating New Knowledge Integration,** where aggregation evolves from static averaging to

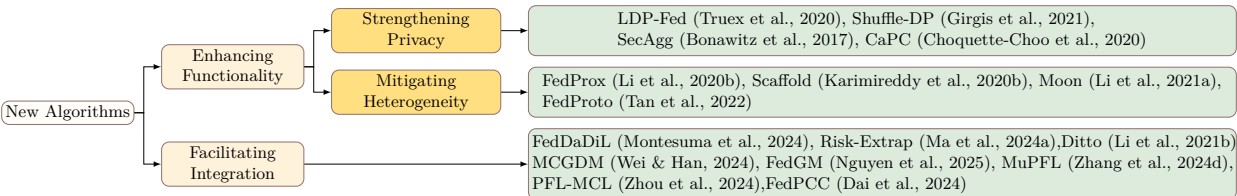

Figure 4: A taxonomy of technical approaches for incorporating new algorithms into FL systems can be organized into two broad categories: (i) methods that enhance existing functionalities, and (ii) methods that facilitate the integration of additional new knowledge. For the former, we consider algorithm switching to strengthen privacy protection and mitigate data heterogeneity.

dynamic, structure-aware mechanisms that help absorb new features, new tasks, and heterogeneous models without destabilizing the federated process. A detailed taxonomy of these technical approaches is illustrated in Figure 4.

## 7.1 Methods

### 7.1.1 For Enhancing Current Functionality

**Mitigating Data Heterogeneity.** A long-standing challenge in FL is that client data are typically non-IID and unevenly distributed, which induces *client drift* and makes naive averaging unstable or slow to converge. To address this, a series of aggregation and local optimization designs explicitly incorporate heterogeneity into the training dynamics. FedProx (Li et al., 2020b) modifies the local objective with a proximal regularization term, constraining local updates from deviating excessively from the current global model, thereby stabilizing training when local objectives differ substantially. Complementarily, Scaffold (Karimireddy et al., 2020b) targets the bias introduced by heterogeneous local updates by using control variates to correct local gradients, encouraging local optimization to better align with the global descent direction. Beyond constraining parameters or gradients, Moon (Li et al., 2021a) emphasizes that heterogeneity can manifest as representation misalignment across clients; it adopts a model-contrastive learning objective to encourage local representations to remain consistent with the global model, reducing over-specialization to local distributions. FedProto (Tan et al., 2022) further explores a more semantic and communication-friendly form of alignment by exchanging class prototypes rather than raw gradients/parameters, enabling clients to align class-wise feature representations under distribution skew. Collectively, these methods demonstrate that enhancing functionality under non-IID data often requires moving from purely averaging-based aggregation to *heterogeneity-aware* optimization and knowledge alignment mechanisms.

**Strengthening Privacy Protection.** In practical FL deployments, maintaining functionality also requires protecting client information throughout training and inference-related exchanges. Two mainstream directions are differential privacy (DP) (Dwork, 2006) and secure Multi-Party Computation (MPC) (Goldreich, 1998). DP-based methods typically perturb client updates (e.g., gradients or parameters) to ensure that each client's contribution is protected within a formal privacy budget. For example, FL with local DP (Truex et al., 2020) performs on-device perturbation before transmission, providing strong protection even under a weak trust assumption about the server, though it may introduce additional noise into optimization. FL with shuffle DP (Girgis et al., 2021) improves the privacy-utility trade-off by anonymizing the association between clients and their updates through shuffling, yielding privacy amplification while often retaining better model utility than purely local perturbation. Recent work further aims to harmonize heterogeneous differential privacy mechanisms and privacy budgets across clients/rounds to improve both accuracy and convergence in DP-FL (Feng et al., 2024a). In parallel, MPC-based approaches provide cryptographic protection by transforming client-side information, including model gradients, intermediate neural representations, and inference predictions, into secrets for secure sharing and computation. SecAgg (Bonawitz et al., 2017) ensures that the server can only recover the aggregated sum (or average) of client updates without inspecting any individual update, making it a foundational primitive for secure aggregation at scale. CaPC (Choquette-Choo et al., 2020) similarly enables confidential collaborative computation by secret-sharing client data for secure

processing, supporting privacy-preserving learning when richer exchanged signals are needed beyond raw parameter updates. Overall, these privacy-preserving techniques enhance current FL functionality by reducing information leakage risks while allowing aggregation to proceed in a principled (DP) or cryptographically protected (MPC) manner.

### 7.1.2 For Facilitating New Knowledge Integration

New aggregation algorithms play a pivotal role in facilitating the integration of new knowledge by evolving from static averaging to dynamic mechanisms that act as the system's control logic. To explicitly address the heterogeneity of new features, aggregation strategies are shifting towards metric-driven and fine-grained alignment. For instance, FedDaDiL (Montesuma et al., 2024) modeled distribution shifts using Wasserstein barycenters to align feature spaces based on optimal transport geometry, while Ma et al. (Ma et al., 2024a) designed a risk extrapolation strategy to optimize aggregation coefficients for causal invariance. At a finer level, Wei & Han (2024) and Nguyen et al. (2025) maximized gradient consistency to reduce distribution discrepancies by matching intra/inter-domain gradients. When new tasks introduce severe functional conflicts, algorithms facilitate integration by dynamically modifying the aggregation topology. PFL-MCL (Zhou et al., 2024) and MuPFL (Zhang et al., 2024d) proposed iterative or adaptive clustering mechanisms that route diverse updates to different "knowledge communities", effectively preventing negative transfer between conflicting tasks. Finally, to accommodate new models or heterogeneous architectures, algorithms are adopting decoupled or parameter-agnostic aggregation schemes. Ditto (Li et al., 2021b) introduced a bi-level mechanism that regularizes local models towards the global average without forcing synchronization, creating an elastic buffer for diverse model parameters. Similarly, FedPCC (Dai et al., 2024) integrated prototype clustering into aggregation to unify feature spaces, allowing the integration of knowledge regardless of the underlying model structural differences.

## 7.2 Real-world Applications

In the healthcare domain, Khan et al. (2025) proposed EAH-FL, an encrypted in-network aggregation framework deployed at the network edge to effectively reduce both communication and computation overhead. It enables efficient federated training under low-bandwidth conditions and substantially improves communication latency and energy consumption without noticeably sacrificing model accuracy, demonstrating strong practicality for real-time medical monitoring. In industrial IoT, Mughal et al. (2024) proposed MEC-AI (HetFL), which adopts a multi-tier edge aggregation architecture with asynchronous updates to reduce communication costs. By orchestrating collaborative training across edge clusters and dynamically selecting clients and allocating resources based on network conditions, it avoids long-haul cloud transmissions and mitigates the straggler effect that can slow down global convergence. In smart agriculture, Li et al. (2024a) proposed a pruning-based aggregation approach that jointly reduces communication and computation costs. The method prunes client models in each round and aggregates the pruned updates, significantly decreasing the transmitted parameters and the local training burden on resource-constrained devices.

## 7.3 Future Works

In this direction, future research should address the seamless integration of novel aggregation algorithms into existing frameworks. First, it remains to be determined whether directly switching aggregation algorithms could slow down the convergence speed of the FL model or impair its final convergence performance, as different algorithms possess distinct optimization trajectories and implicit regularization effects (Lim et al., 2025) that may lead to transient instability or divergence when the aggregation rule is abruptly altered. Second, if such a switch is viable, identifying the optimal timing for this transition is critical, requiring research to determine if specific training phases exist where switching algorithms minimizes disruption while maximizing convergence gains and how to determine this timing systematically (Karimireddy et al., 2020a). Third, significant technical difficulties arise when the new algorithm requires the communication of additional historical information that was not stored during the training process of the old algorithm, necessitating methods to approximate missing state variables, such as momentum buffers or control variates to enable a smooth transition between stateless and stateful algorithms (Kiani et al., 2025). Finally, the aggregation mechanism must be capable of handling scenarios where the model utility and privacy requirements of different clients

are distinct at both spatial and temporal levels (Su et al., 2023), developing adaptive strategies that can dynamically reconcile these heterogeneous and evolving constraints remains an open problem for the next generation of privacy-preserving federated systems.

# 8 Future Works of FL with New Knowledge

## 8.1 Future Tendency and Open Problems

### 8.1.1 Future Tendency

FL is increasingly moving from *one-off training* to *lifelong system evolution*, where an established system must continuously incorporate *new knowledge* to satisfy new demands while preserving existing functionality. In our setting, new knowledge can arise from four sources: **new features, new tasks, new models, and new algorithms**, and may be introduced by dynamically changing client groups (clients with only old knowledge, only new knowledge, or mixed knowledge). Such continual arrivals create time-varying objectives, higher system dynamics, and tighter resource constraints, especially in cross-device FL.

Future FL research will increasingly treat *sustainable upgrades* as explicit system-level objectives, rather than side effects of improving optimization in a fixed setting. This implies a shift from "improving a fixed task/model" to "maintaining a living system" that supports recurrent upgrades with bounded cost and bounded regressions. Practically, this calls for versioned models/algorithms, upgrade policies, and rollback-ready deployment pipelines, making evolutionary FL closer to an MLOps-like continual delivery process. In parallel, as model technology advances, new architectures (e.g., transformers and foundation models) are likely to serve as *universal interfaces* for absorbing diverse new knowledge, which encourages modular designs (adapters, prompt modules, task heads, split components) to enable parameter-efficient upgrades, partial activation, and backward compatibility under strict device and bandwidth constraints.

Moreover, aggregation will evolve from static averaging to *dynamic, structure-aware coordination* that explicitly handles heterogeneity in knowledge states, model structures, and client availability. In particular, algorithm evolution (new optimizers, new privacy mechanisms, new robustness designs) will be coupled with the integration of new features/tasks/models, forming a unified coordination layer that decides *who trains what, when, and how*. As continual integration becomes the norm, governance primitives will also become indispensable: detecting whether incoming changes are truly new knowledge (vs. noise or transient drift), validating benefit under partial observability, attributing gains/losses to knowledge sources, and maintaining a reproducible upgrade history. Consequently, benchmarks will shift from single-stage performance to long-horizon outcomes that measure adaptation speed, retention of prior capabilities, cumulative utility over time, resource overhead, and robustness under non-stationarity, with protocols that simulate diverse arrival patterns (form and timing), dynamic client participation, and realistic system constraints.

### 8.1.2 Open Problems

Despite rapid progress, several fundamental problems remain open and will likely define the next phase of FL with new knowledge. First, we still lack principled formulations and guarantees for *non-stationary evolution*: it remains unclear how to mathematically define "successful evolution" when objectives shift over time, and how to unify fast adaptation to new knowledge, bounded forgetting of old knowledge, and constraints on communication/energy/latency into a single optimization target with stability and convergence guarantees under dynamic participation. Closely related is the challenge of *detecting and characterizing new knowledge* in realistic heterogeneous environments: when new features or tasks appear, the system must decide whether to adapt, ignore, or defer, yet reliable novelty detection without labels is difficult, and one must distinguish genuine new demands from transient fluctuations, device faults, or low-quality data while controlling false alarms and missed detections.

Second, evolutionary FL must support *regression-free integration* across heterogeneous client knowledge states (only old, only new, or mixed). Updates that benefit new-knowledge clients can harm old-knowledge clients, raising open questions on how to enforce "no significant harm" constraints during upgrades, how

to design personalization that remains compatible with future global evolution, and how to quantify and control long-term fairness and accessibility as the system evolves.

Finally, we need *dependency-aware co-evolution* and *safe switching* mechanisms at the system level. New knowledge types often co-occur and depend on each other (e.g., a new task may require a representational upgrade), but we lack robust methods to build dependency graphs and plan integration sequences under resource budgets. Meanwhile, switching aggregation algorithms or model architectures may require additional states (e.g., control variates, momentum, masks, prompts) that are missing or inconsistent across clients, making state migration, compatibility layers, and rollback/canary mechanisms under federated constraints key unresolved issues. In addition, the community still lacks benchmarking standards for simulating arrival form/timing, dynamic client sets, and long-horizon evaluation; beyond final accuracy, metrics should capture adaptation speed, forgetting, cumulative utility, stability under upgrades, and system costs to enable reproducible progress and meaningful comparisons.

Table 9: A synthesis of the threats, typical defenses, and the trade-offs that should be considered when incorporating each type of new knowledge into FL.

| Knowledge Type | Threat | Defense | Trade-off |
|---|---|---|---|
| New features | Feature/gradient leakage; trigger injection | Sanitization; DP + secure aggregation; anomaly checks | Lower utility; more compute/comm |
| New tasks | Task inference; negative transfer; head backdoors | Task isolation (heads/adapters); robust agg.; DP/SA | Less sharing; larger models; coordination overhead |
| New models | Trojaned weights; inversion/leakage; supply-chain risk | Provenance/attestation; backdoor scans; DP/SA; sandbox eval | Screening cost; slower training; accuracy loss (DP) |
| New algorithms | Weaker privacy guarantees; poisoning/Byzantine | Privacy accounting + DP; Byzantine-robust agg.; SA/MPC | Robustness–accuracy; extra rounds; tuning complexity |

## 8.2   Threats

Incorporating new knowledge into an FL system often requires communicating additional information beyond standard model parameters or gradients. Therefore, it is urgent to propose dedicated methods to protect these additional contents in the semi-honest scenarios, in particular, against a variety of inference attacks (Wang et al., 2019; Luo et al., 2021). From a representation perspective, Soteria provided a provable defense by perturbing privacy-sensitive representation components to reduce reconstruction leakage while preserving utility (Sun et al., 2021). Complementarily, task-agnostic privacy-preserving representation learning has been proposed to mitigate attribute inference attacks in FL without assuming a specific downstream task (Arevalo et al., 2024). Some preliminary tries include adding random (Dong et al., 2022) or DP noise (Zizzo et al., 2022) to the domain or task-specific models or class prototypes of FCL. However, such noise-based protection often comes at the cost of model performance. Given FL's multi-party nature, leveraging cryptographic methods like MPC is a promising alternative, as it can provide strong security guarantees without sacrificing utility. However, such privacy protection hurts the performance of FL systems. Aligning with the intrinsic nature of multiple parties in FL, it may be possible to leverage MPC to communicate the additional contents. MPC is built on cryptography, and there are strong security guarantees. Besides, because the carriers of new knowledge are features, tasks, models, and algorithms, we need to consider the worst case that these carriers are polluted or poisoned in malicious scenarios. In this context, FedTilt studied robust FL under persistent outliers while incorporating multi-level fairness constraints across clients and groups (Zhang et al., 2025b). Provenance primitives such as watermarking can further support ownership verification and model provenance tracking; for example, FedGMark provided a certifiably robust watermarking approach for federated graph learning (Yang et al., 2024c). For instance, features with poison attacks may be disguised as new features, and if there is no detection tool, they will be naturally incorporated into FL. An optimization-based attack framework that broke state-of-the-art poisoning defenses in FL was presented in (Yang et al., 2024e). Furthermore, malicious clients may conduct unintended local model training for society-harmful tasks; in this case, there is a need to distinguish them from new benign tasks. New models may also contain backdoors (Xie et al., 2020), blindly incorporating them will cause serious outcomes. A secret sharing–inspired robust distributed backdoor attack against FL was studied in (Yang et al., 2025). Distributed backdoor attacks on federated graph learning and certified defenses were investigated in (Yang et al., 2024d). Finally, new algorithms themselves can introduce new attack surfaces; for example, some gradient inversion attacks (Huang et al., 2021) are only effective against specific FL algorithms. In summary, we provide a synthesis of the threats, typical defenses, and the trade-offs that should be considered for each type of new knowledge, as shown in Table 9.

### 8.3 Incentive

To build a continuously evolving FL ecosystem, the core is designing a dynamic incentive mechanism that effectively manages the knowledge lifecycle. This mechanism must address three key issues: how to fairly evaluate the contributions of different types of knowledge, how to incentivize the addition of high-quality knowledge, and how to handle the secure exit of knowledge.

Fair quantification of contributions is the cornerstone of all incentive mechanisms. Within the new knowledge framework defined in this paper, we must go beyond traditional evaluation methods that focus solely on data volume (Li et al., 2020a) and establish a multi-dimensional model capable of measuring the value of heterogeneous knowledge. Table 10 summarizes a compact instantiation of this model across the four carriers. For contributions of new features, the Shapley value method can be employed. Guo et al. (2024a) state that when a participant introduces a new feature, their contribution can be measured by the marginal improvement of this feature on the global model's performance. For contributions of new tasks, the net value evaluation method (Kirkpatrick et al., 2017) is utilized. When a new task is introduced, it brings in new knowledge but may also interfere with existing tasks, causing catastrophic forgetting; thus, its contribution must be a net effect. For contributions of new models or algorithms, a multi-dimensional weighted gain method can be adopted, comprehensively considering communication efficiency (Chen et al., 2023a), computational efficiency (Pan et al., 2024), and model performance (Nilsson & Smith, 2018).

After being able to measure contributions, it's crucial to incentivize participants with high-quality knowledge to contribute honestly. Since participants have private information about the true quality of their contributions, contract theory (Liu et al., 2022) offers a suitable solution. Instead of offering a fixed price, a "contract menu" can be designed for participants to choose from. Each menu item pairs a commitment level with a corresponding reward. Importantly, the reward for each contract must exceed the cost incurred by participants in contributing their knowledge; otherwise, participation will be discouraged. Meanwhile, the reward gap should be carefully designed so that participants with high-quality knowledge maximize benefits only by selecting high-commitment, high-return contracts. This encourages them to "tell the truth," thus attracting high-quality knowledge into the system.

A healthy ecosystem must allow participants to withdraw their contributions safely when needed. Federated unlearning (Zhao et al., 2025b) is central to achieving this. When a participant requests to withdraw their knowledge (whether features, tasks, or models), the system must perform an unlearning operation, which incurs unlearning costs. Such unlearning requests may be proposed by different participants at different times, thus preserving normal model functionality while achieving effective continuous unlearning (Gao et al.) is required. This cost depends on the thoroughness of unlearning, categorized into approximate unlearning (Xiong et al., 2024) and exact unlearning (Qiu et al., 2023). Approximate unlearning efficiently reduces the influence of specific knowledge with lower computational costs and faster speed. Exact unlearning usually requires partial or complete retraining from a checkpoint, excluding the target knowledge, leading to higher computational costs. The unlearning cost forms the basis for designing the exit clause in the incentive mechanism. Within this framework, the exit clause can explicitly link the unlearning cost to the participant's responsibilities. Participants can choose between approximate or exact unlearning based on their needs. Choosing the former may involve a small fee or a deduction of reputation points, while the latter entails bearing a higher proportion of computational costs. Additionally, for long-term contributors with high reputation scores, the system can subsidize part of the unlearning cost when they exit, recognizing their past contributions.

Table 10: A compact multi-dimensional model for valuing contributions of new knowledge in evolving FL.

| Carrier | What changes | Contribution |
| --- | --- | --- |
| New features | Input distribution / modality | Marginal utility (e.g., Shapley-style attribution) and robustness to feature shift |
| New tasks | Task set / functionality | Net utility: gain on new tasks minus loss (forgetting) on prior tasks |
| New models | Architecture / capacity | Weighted gain over accuracy, retention, compute, and communication cost |
| New algorithms | Aggregation protocol | Weighted gain over convergence, robustness, privacy; plus transition cost |

# 9 Conclusion

In this paper, we focus on a comprehensive review of studies relevant to incorporating new knowledge into FL to achieve its sustainable development. We consider new knowledge from four sources: features, tasks, models, and algorithms. For incorporating new features, we discuss a set of processes to enhance the out-of-distribution detection and generalization ability of FL systems. Regarding new tasks, we discuss how to use task-personalized and self-supervised FL to strengthen the cross-task generalization capability of FL, and we also introduce how to learn new tasks through federated continual learning. Additionally, we review existing research that may facilitate the fusion of different FL models and the transition between different algorithms. Finally, we comprehensively discuss future research directions, considering factors such as scenario setups, security and privacy threats, and incentives.

## Acknowledgments

We thank the meta and anonymous reviewers for their valuable comments. This research is supported by the National Research Foundation, Singapore, and the Cyber Security Agency of Singapore under the National Cybersecurity R&D Programme and the CyberSG R&D Programme Office (Award CRPO-GC3-NTU-001). Any opinions, findings, conclusions, or recommendations expressed in these materials are those of the author(s) and do not reflect the views of the National Research Foundation, Singapore, the Cyber Security Agency of Singapore, or the CyberSG R&D Programme Office.

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
