# OpenReview forum: "Incorporating New Knowledge into Federated Learning: Advances, Insights, and Future Directions"
_TMLR — Accepted by TMLR_

### Review · Reviewer_cghH · 2025-12-17

**Summary Of Contributions:**

This paper presents a comprehensive survey on incorporating knowledge into federated learning frameworks, a natural and increasingly important area given that clients typically receive streaming data. The authors systematically categorize existing approaches from multiple perspectives, including tasks, features, models, and algorithms. In addition, the survey highlights open challenges and discusses potential directions for future research.

**Strengths**
- The paper is well-structured and easy to follow.
- The paper covers a wide range of literature on incorporating new knowledge — from adaptation and generalization to personalization and continual learning — going beyond existing surveys that usually focus on only one aspect.
- Given the growing interest in continual learning, the survey is timely.

**Weaknesses**
- The paper provides limited comparison and analysis across different methods, making it difficult to understand the current progress of the field and the stage it has reached.
- The discussion of real-world feasibility is limited. While several real-world applications are mentioned, most cited works are still evaluated in relatively controlled research settings. Important real-world complexities, such as synchronization issues and system/data heterogeneity, are not sufficiently discussed.

**Additional Comments:**

N/A

**Audience:**

Yes

**Audience Explanation:**

Given the growing interest in continual learning across the ML community, as well as the streaming-data nature of federated learning, this paper is likely to be of interest to a broad audience.

Also, by covering a wide range of related areas, the paper can attract readers from different subfields and help them understand the current status of research in this space.

**Claims And Evidence:**

Yes

**Claims Explanation:**

The paper provides a well-organized categorization and, to the best of my knowledge, covers all major topics related to incorporating new knowledge. Each category is supported by a substantial body of relevant literature, which supports the claims made in the paper.

**Requested Changes:**

Building on the weaknesses mentioned above, I suggest the following improvements:

- The authors could provide benchmark comparisons across representative methods in each category, or at least point to existing benchmark studies, to help readers better understand the relative strengths and limitations of different approaches.
- A deeper discussion on the gap between research and practice would be valuable, particularly in the context of **continual learning–oriented federated learning**, rather than standard static FL settings. Discussing challenges *unique* to continual and streaming scenarios would further strengthen the paper. Related perspectives from prior work (e.g., Daly et al.; Kuo et al.) could serve as useful references.

[1] Daly et al. Federated Learning in Practice: Reflections and Projections

[2] Kuo et al. Research in Collaborative Learning Does Not Serve Cross-Silo Federated Learning in Practice

---

> ### Author Response · Authors · 2026-01-14
> **Response to Reviewer cghH**
>
> We thank the reviewer for noting the structure, breadth, and timeliness. We agree with the main weaknesses: insufficient cross-method comparison/analysis and limited discussion of real-world feasibility.
>
> > The paper provides limited comparison and analysis across different methods, making it difficult to understand the current progress of the field and the stage it has reached.
>
> In our current revision, we include a detailed comparison (Tables 2, 4-8) of different methods in terms of the timing they support, the duration for which the knowledge remains available, the overhead they incur (including computation, communication, and storage costs), and the performance they can achieve. We also summarize the methodological trends for each knowledge type, and provide an overall outlook on future directions for FL with new knowledge in the Future Work section.
>
>
>
> > The discussion of real-world feasibility is limited. While several real-world applications are mentioned, most cited works are still evaluated in relatively controlled research settings. Important real-world complexities, such as synchronization issues and system/data heterogeneity, are not sufficiently discussed.
>
> We currently motivate dynamic deployment realities in the introduction and focus primarily on cross-device settings given their higher dynamics and constraints, but we should explicitly discuss practice-driven constraints (asynchrony, stragglers, dropouts, heterogeneous hardware, nonstationary participation, delayed labels). We will make the following revisions.
>
> Planned revision:
> - Add a "research vs practice" subsection that itemizes practical deployment constraints and explains how they impact each of the four new-knowledge carriers.
> - Incorporate additional practice-oriented references suggested by the reviewer (e.g., Daly et al.; Kuo et al.) into that subsection and connect them to our scenario axes + decision framework.
> - Add a synchronization/asynchrony axis explicitly into the scenario framework and show how it changes method choice and evaluation design.

---

> > ### Comment · Reviewer_cghH · 2026-01-31
> >
> > I appreciate the authors’ detailed response. I have reviewed the revised manuscript as well as the additional clarifications provided in response to other reviewers’ comments. The revisions have largely addressed my concerns, and I am therefore inclined toward acceptance.
> >
> > I have a few minor suggestions for further improvement:
> >
> > - In the newly added tables, it would be helpful to briefly explain terms such as timing support, period support, abrupt, and graduate directly in the table captions. This would make the tables more self-contained and reduce the need for readers to search for definitions in the main text.
> >
> > - The planned revisions for "research vs. practice" outlined by the authors sound reasonable, and I look forward to seeing the updated version.

---

### Review · Reviewer_4jP1 · 2025-12-31

**Summary Of Contributions:**

### Summary

This paper surveys how deployed federated learning systems can **incorporate “new knowledge”** to extend their **lifespan**, organizing the literature into four sources of new knowledge: **new features, new tasks, new models, and new aggregation algorithms**, with sections on feature shift pipelines (FDG/FODD/FDA), task generalization/personalization, federated continual learning, model transitions (e.g., foundation models/LLMs), and security/incentives.

### Strengths

* **Timely and relevant** framing around evolving FL systems rather than static FL.
* **Broad coverage** with structured taxonomies/figures that help navigation.
* Includes discussion of **threats and incentives/unlearning**, which is often missing in purely algorithmic surveys.

### Weaknesses

* Currently reads more like a **well-organized catalog** than a survey that **creates new connections / highlights trends / sharpens open problems** (the bar stated in the TMLR survey guidance).
* The claimed focus on **arrival timing/form** and **lifespan** is not yet made **operational** (few scenario axes or “what to use when” guidance).
* Security is discussed, but not well integrated into the earlier taxonomy (missing “new-knowledge type × threat × defense tradeoff” synthesis).

**Audience:**

Yes

**Audience Explanation:**

**Yes.** A meaningful portion of TMLR’s audience working on federated learning, distributed/edge ML, continual learning, personalization, and deployment robustness would likely find this survey useful—especially because it frames FL as an **evolving system** and tries to unify several subareas (feature shift/adaptation, new tasks, model transitions, and aggregation changes) under a single “new knowledge” lens.

That said, the *level* of interest will depend on revisions: the paper will be substantially more compelling to TMLR readers if it strengthens its **synthesis** (new cross-area connections, trend summaries, and clearly stated open problems) beyond method cataloging.

**Claims And Evidence:**

Yes

**Claims Explanation:**

**Partially.** The submission provides a broad literature-backed narrative and many cited examples that plausibly support the high-level taxonomy (“new features / tasks / models / algorithms”) and the motivation that FL deployments must evolve over time.

However, several *central* claims are **not supported with sufficiently clear or convincing evidence** in their current form:

* **“Lifespan” as a core objective**: the paper introduces a lifespan notion (e.g., based on client count dropping to a threshold), but does not substantiate why this is the right operational definition or how it correlates with real system utility/service quality; nor does it provide evidence that surveyed methods extend this lifespan under consistent metrics.
* **“Arrival timing/form of new knowledge” analysis**: the paper states this as a key contribution, but the evidence is mostly descriptive; it lacks explicit scenario axes, empirical comparisons, or a decision framework showing that timing/form materially changes which methods work.
* **Connections and trends**: many sections list methods, but the paper rarely backs “trend” statements with aggregated evidence (e.g., comparative tables, consensus findings, or clearly stated open problems grounded in recurring limitations).

**Overall rating for this question:** *Moderate support*. The survey is credible as a reference map, but the strongest “survey-style” claims (new synthesis, trends, and operational guidance) need clearer evidence and more explicit consolidation to be fully convincing.

**Requested Changes:**

### Requested changes (high impact)

1. Add 2–4 explicit **new cross-area connections** (design patterns) that tie the four categories together, each with takeaways.
2. Make **timing/form** concrete via scenario axes (abrupt/gradual, sync/async, churn, labeled/unlabeled) + a decision table mapping scenarios → method families → failure modes/costs.
3. Strengthen **benchmark/metrics synthesis** (which datasets map to which “new knowledge” settings; what common metrics miss).
4. Integrate security with a compact matrix: **knowledge type × threat × typical defenses × tradeoffs**.

---

> ### Author Response · Authors · 2026-01-14
> **Response to Reviewer 4jP1 [Part 1]**
>
> We thank the reviewer for the careful and constructive assessments. We agree that the current draft is stronger as a structured reference map than as an operational, evidence-backed synthesis. Below we respond point-by-point to the requested high-impact changes that are sammary of your indicated weaknesses and the evidence-related concerns about central claims.
>
> > Add 2–4 explicit new cross-area connections (design patterns) that tie the four categories together, each with takeaways
>
> While our current draft provides a structured map and taxonomy, we can strengthen "survey value-add" by extracting cross-cutting patterns and summarizing recurring limitations across subareas. First, Section 5 ("Incorporate New Tasks with New Features") highlights a representative cross-area connection between new features and new tasks. It also points to several related directions that build on, and are inspired by, methods for jointly incorporating new features and tasks. For example, FCL may need to detect the arrival of new tasks by identifying sharp shifts in the feature distribution, and it may also benefit from self-supervised learning to enable more efficient learning of new tasks.
> Besides, in our current revision, we have restructured Sections 6 and 7 to explicitly formalize cross-area connections, moving beyond isolated categories to demonstrate system-level synergy. Specifically, we added two new subsections organized by the specific type of heterogeneity (Features, Tasks, and Models) they address:
>
> - Section 6.1.2 (Methods For Facilitating New Knowledge Integration): We formalize the "Model-as-Interface" design pattern. This section details how new architectures serve as flexible infrastructure. We structure this discussion by:
>   - For New Features: Using auxiliary components like global adapters and generative modules to bridge distribution gaps.
>   - For New Tasks: Using modular structures like prompt pools, architecture decomposition, and hypernetworks to manage sequential knowledge without interference.
>
> - Section 7.1.2 (Methods For Facilitating New Knowledge Integration): We formalize the "Algorithm-as-Controller" design pattern. This section demonstrates how aggregation algorithms evolve into state-aware control logic. We categorize these mechanisms into three levels:
>   - For New Features: Employing metric-driven (e.g., Wasserstein, MMD) and gradient-alignment strategies to handle distribution shifts.
>   - For New Tasks: Utilizing dynamic clustering and routing mechanisms to modify aggregation topology and resolve functional conflicts.
>   - For New Models: Adopting decoupled (e.g., bi-level optimization) or prototype-based aggregation to accommodate heterogeneous model architectures.
>
> These additions explicitly tie the four categories together: "Features" and "Tasks" represent the challenges, while "Models" and "Algorithms" serve as the enablers that facilitate their integration.
>
> Finally, we have included a new subsection (Section 8.1) in the last section, expecting the future tendency and open problems for FL with new knowledge from a high level. Specifically, we expect that FL is shifting from one-off training to a continually evolving deployed system that must absorb new features, tasks, models, and algorithms while preserving existing behavior under tight device and participation constraints. This raises open problems in defining and guaranteeing non-stationary evolution, detecting true new knowledge without labels, integrating updates without regressions for old knowledge clients, and enabling dependency-aware upgrades with safe switching and long-horizon benchmarks that measure adaptation speed, retention, stability, and cost.
>
>
> > Make timing/form concrete via scenario axes (abrupt/gradual, sync/async, churn, labeled/unlabeled) + a decision table mapping scenarios → method families → failure modes/costs.
>
> We acknowledge that defining the lifespan of FL based solely on the number of existing clients is neither operational nor logically sound, since many factors can influence when clients join or leave. The same concern applies to the notion of a "usable lifespan" for FL systems. Therefore, in our current revision, we instead use the scenario where current demands must be met in a timely manner to motivate incorporating new knowledge into existing FL systems. Under this framing, we do not need to explicitly define an FL lifespan. As long as client demands can be satisfied, the FL system remains usable. Regarding the timing and form of new knowledge, our revision includes a detailed comparison (Tables 2, 4-8) across methods in terms of the timing they support, the duration over which the knowledge remains available, the overhead they incur (including computation, communication, and storage costs), and the performance they can achieve. In future revisions, we also plan to add a decision table mapping scenario → candidate method families from 4 knowledge types → expected failure modes/costs.

---

> ### Author Response · Authors · 2026-01-14
> **Response to Reviewer 4jP1 [Part 2]**
>
> > Strengthen benchmark/metrics synthesis (which datasets map to which “new knowledge” settings; what common metrics miss).
>
> We currently provide Table 3 listing commonly used datasets, but we do not yet map datasets to scenario axes nor critique metric gaps. In our current revision, we have included a figure (Figure 2) to illustrate the dataset-to-setting mapping relation. In the future revision, we will also add a metric gap summary (e.g., average accuracy vs worst-domain accuracy; continual learning forgetting measures; system-level latency/time-to-adapt; privacy/robustness tradeoffs).
>
>
> > Integrate security with a compact matrix: knowledge type × threat × typical defenses × tradeoffs.
>
>
> Currently, we acknowledge trusted-settings assumptions in the main body and discuss threats later (e.g., poisoned features, harmful tasks, backdoored models, algorithm-specific vulnerabilities), but the integration into the four-type taxonomy is not yet systematic. In the current revision, we have included a matrix (Table 9) in the form of knowledge type × threat × typical defenses × tradeoffs and then reference it within each earlier section (feature/task/model/algorithm) as a "security sidebar" so security becomes a first-class axis rather than a late add-on.

---

> > ### Comment · Reviewer_4jP1 · 2026-02-19
> > **Thanks for the response**
> >
> > I am happy with the authors' rebuttal. I vote for acceptance.

---

### Review · Reviewer_xne5 · 2026-01-01

**Summary Of Contributions:**

**Strengths**
- The focus on FL system evolution and lifespan is timely and addresses a critical gap between rapid ML advancements and the high cost of training FL models from scratch.
- The 4-category framework (Features, Tasks, Models, Algorithms) is logical and provides a clear structure for reviewing a large body of work.
- The paper reviews recent literature (up to 2025) across multiple sub-fields within FL.
- Future work on threats and incentives are valuable for guiding future research.



**Improvement / Clarifications**

1. Positioning & Contribution Clarity: The abstract and introduction clearly state the problem. However, the novelty claim as the "first" survey to address FL evolution could be strengthened by more sharply contrasting the proposed evolutionary framework with the "static" assumptions of prior surveys mentioned in Table 1.
2. Structural Flow: Section 3 (New Features) is very detailed and well-organized. The transition to Section 7 (New Algorithms) feels somewhat abrupt and less developed compared to previous sections. It raises excellent questions but lacks a review of existing work that attempts such transitions. Expanding this section would improve balance.
3. Technical Depth vs. Readability: The draft is highly technical and assumes significant reader expertise in FL. While appropriate for a survey, adding more intuitive explanations of core challenges (e.g., why non-IID data makes FCL so hard) in early sections would improve accessibility.
4. Figures and Visualizations: The referenced figures (Fig 1, 2, 3, 4, 5) are crucial for understanding the taxonomies and workflows. Ensure these are clear, well-labeled, and directly referenced in the text. Their captions should be self-explanatory.
5. Future Directions (Section 8): The discussion on Incentives is insightful, linking contribution evaluation to the knowledge taxonomy. Consider adding a brief diagram or table to summarize the proposed "multi-dimensional model" for quantifying contributions of features, tasks, and models. The Threats subsection rightly identifies new attack surfaces. It could briefly mention the tension between proactive detection of malicious new knowledge (e.g., poisoned features) and the privacy of client data.

**Audience:**

Yes

**Audience Explanation:**

This survey paper systematically reviews the challenge of integrating new knowledge—such as new features, tasks, models, and algorithms—into existing Federated Learning (FL) systems to enable sustainable evolution and extended lifespan without discarding prior investments.

The paper positions itself as the first survey to examine FL literature through the lens of system evolution and lifespan extension due to new knowledge integration, differentiating it from prior surveys focused on static settings or isolated techniques.

**Claims And Evidence:**

No

**Claims Explanation:**

## Missing Important Relevant References

**FL for Healthcare**

Pang et al. A Tri-Factor Adaptive Federated Learning Framework for Parkinson's Disease Diagnosis via Multi-Source Facial Expression Analysis. IEEE Journal of Biomedical and Health Informatics, 2025

Pang et al. Breaking Data Silos in Parkinson’s Disease Diagnosis: An Adaptive Federated Learning Approach for Privacy-Preserving Facial Expression Analysis. AAAI Conference on Artificial Intelligence, 2025

**FL Algorithm for Graph Data**

Wang et al. GraphFL: A Federated Learning Framework for Semi-Supervised Node Classification on Graphs. IEEE International Conference on Data Mining, 2022

Yang et al. FedGMark: Certifiably Robust Watermarking for Federated Graph Learning. Neural Information Processing Systems, 2024

**Security of FL**

Yang et al. A Secret Sharing-Inspired Robust Distributed Backdoor Attack to Federated Learning. ACM Transactions on Privacy and Security, 2025

Zhang et al. FedTilt: Towards Multi-Level Fairness-Preserving and Robust Federated Learning. Deep Learning Security and Privacy Workshop, 2025

Yang et al. Distributed Backdoor Attacks on Federated Graph Learning and Certified Defenses. ACM Conference on Computer and Communications Security, 2024

Yang et al. Breaking State-of-the-Art Poisoning Defenses to Federated Learning: An Optimization-Based Attack Framework. ACM International Conference on Information and Knowledge Management, 2024

**Privacy of FL**

Feng et al. Universally Harmonizing Differential Privacy Mechanisms for Federated Learning: Boosting Accuracy and Convergence. ACM Conference on Data and Application Security and Privacy, 2025

Arevalo et al. Task-Agnostic Privacy-Preserving Representation Learning for Federated Learning Against Attribute Inference Attacks. AAAI Conference on Artificial Intelligence, 2024

Sun et al. Soteria: Provable Defense against Privacy Leakage in Federated Learning from Representation Perspective. IEEE/CVF Conference on Computer Vision and Pattern Recognition, 2021

**Requested Changes:**

See Improvement / Clarification

---

> ### Author Response · Authors · 2026-01-14
> **Response to Reviewer xne5 [Part 1]**
>
> We sincerely thank the reviewer for the positive assessment of the paper’s timeliness and the clarity of our 4-category framework. We appreciate the concrete suggestions; they will materially improve the paper’s positioning, balance, and readability.
>
> > Positioning & Contribution Clarity: The abstract and introduction clearly state the problem. However, the novelty claim as the "first" survey to address FL evolution could be strengthened by more sharply contrasting the proposed evolutionary framework with the "static" assumptions of prior surveys mentioned in Table 1.
>
> While our introduction already highlights that "current FL systems typically assume a fixed and predetermined distribution of data and tasks" and Table 1 is intended to contrast our lens with prior surveys, we agree that the difference is not yet crisp enough, and the "first" claim should be more carefully supported. Therefore, in the current phase, we have made the following revisions:
>
>
>
> In abstract, we add:
>
> - Unlike prior surveys that primarily catalogue FL techniques under a fixed system specification, we adopt a lifecycle evolution perspective and synthesize methods that enable time-varying integration of new features, tasks, models, and aggregation algorithms while preserving existing functionality.
>
>
> In the introduction, we add a bridge for Table 1, and highlight the lifecycle perspective of our survey:
>
> - Prior surveys largely provide a snapshot taxonomy of FL methods under a fixed system specification, where the feature/task space, model architecture, and aggregation protocol are treated as predetermined, and heterogeneity is addressed within this static pipeline. Consequently, they seldom discuss the system's lifecycle questions that arise in deployments, including when new knowledge should trigger an upgrade, how the upgrade should be realized (e.g., detection, adaptation, or migration), and how to bound regressions on previously supported functionality. In this survey, we adopt a lifecycle evolution perspective and synthesize upgrade mechanisms along four evolving variables, including features, tasks, models, and aggregation algorithms, while highlighting how the arrival form and timing of new knowledge affect the incorporation process.
>
> > Structural Flow: Section 3 (New Features) is very detailed and well-organized. The transition to Section 7 (New Algorithms) feels somewhat abrupt and less developed compared to previous sections. It raises excellent questions but lacks a review of existing work that attempts such transitions. Expanding this section would improve balance.
>
>
>
> In the current manuscript, Section 7 is comparatively short and mainly introduces two buckets (heterogeneity-handling vs privacy-protecting aggregation), which can feel like a jump after the deeper taxonomies earlier (e.g., feature-side workflow and datasets).
>
> In our current revision, we have restructured Section 7 to explicitly formalize cross-area connections, moving beyond isolated categories to demonstrate system-level synergy. Specifically, we added two new subsections organized by the specific type of heterogeneity (Features, Tasks, and Models) they address:
>
> - Section 7.1.2 (Methods For Facilitating New Knowledge Integration): We formalize the "Algorithm-as-Controller" design pattern. This section demonstrates how aggregation algorithms evolve into state-aware control logic. We categorize these mechanisms into three levels:
>   - For New Features: Employing metric-driven (e.g., Wasserstein, MMD) and gradient-alignment strategies to handle distribution shifts.
>   - For New Tasks: Utilizing dynamic clustering and routing mechanisms to modify aggregation topology and resolve functional conflicts.
>   - For New Models: Adopting decoupled (e.g., bi-level optimization) or prototype-based aggregation to accommodate heterogeneous model architectures.
>
> These additions explicitly tie the four categories together: "Features" and "Tasks" represent the challenges, while "Models" and "Algorithms" serve as the enablers that facilitate their integration.
>
>
> After Section 6, we add one short paragraph to bridge Section 7:
>
> - Model evolution expands the capacity and interface of an FL system, but system-level evolution also depends on the coordination logic that turns heterogeneous client updates into global progress. As client populations, data heterogeneity, and privacy constraints change over time, an established FL system may need to upgrade its aggregation rule to preserve existing functionality while absorbing newly arriving knowledge. We therefore next review aggregation algorithms as a fourth carrier of new knowledge in evolving FL systems.

---

> ### Author Response · Authors · 2026-01-14
> **Response to Reviewer xne5 [Part 2]**
>
> > Technical Depth vs. Readability: The draft is highly technical and assumes significant reader expertise in FL. While appropriate for a survey, adding more intuitive explanations of core challenges (e.g., why non-IID data makes FCL so hard) in early sections would improve accessibility.
>
> While Section 2 defines non-IID across clients and frames lifecycle/lifespan, we can add more accessible intuition, especially for continual/streaming settings. In the introduction of the current revision, we add explaination to the non-IID scenario to improve the readability:
>
> - Specifically, in FL, each client optimizes the model using its own data. When client data are non-IID, the updates computed on different devices are optimized for different local distributions, so they can point in inconsistent directions: an update that helps one client may hurt others. As a result, simple averaging is no longer equivalent to taking a clean descent step on a single shared objective, which can slow convergence and reduce stability.
>
> > Figures and Visualizations: The referenced figures (Fig 1, 2, 3, 4, 5) are crucial for understanding the taxonomies and workflows. Ensure these are clear, well-labeled, and directly referenced in the text. Their captions should be self-explanatory.
>
> Figure 1 already provides the high-level lifecycle storyboard, but we agree that the paper should be more robust to "figure-only" reading. Therefore, we have revised all figure captions to be self-contained (define abbreviations, state what the figure is used for, and provide a one-sentence "takeaway").
>
> > Future Directions (Section 8): The discussion on Incentives is insightful, linking contribution evaluation to the knowledge taxonomy. Consider adding a brief diagram or table to summarize the proposed "multi-dimensional model" for quantifying contributions of features, tasks, and models. The Threats subsection rightly identifies new attack surfaces. It could briefly mention the tension between proactive detection of malicious new knowledge (e.g., poisoned features) and the privacy of client data.
>
>
> In the current Section 8.2, we already define a "multi-dimensional model" framing that distinguishes feature/task/model(or algorithm) contributions, but it is described only in text. In Section 8.1, we already noted the worst-case where carriers of new knowledge can be polluted (poisoned features, harmful tasks, backdoored models, algorithm-specific attack surfaces). In the current phase, we have added a Table in section 8.3, which summarizes a compact instantiation of this model across the four carriers:
>
> Table: A compact multi-dimensional model for valuing contributions of new knowledge in evolving FL.
>
> | Carrier        | What changes                    | Contribution                                                                 |
> |---------------|----------------------------------|------------------------------------------------------------------------------|
> | New features  | Input distribution / modality | Marginal utility (e.g., Shapley-style attribution) and robustness to feature shift |
> | New tasks     | Task set / functionality        | Net utility: gain on new tasks minus loss (forgetting) on prior tasks         |
> | New models    | Architecture / capacity         | Weighted gain over accuracy, retention, compute, and communication cost       |
> | New algorithms| Aggregation protocol            | Weighted gain over convergence, robustness, privacy; plus transition cost     |
>
> In future revisions, we also plan to make the following revisions.
>
> Planned revision:
> - Add a paragraph in Threats explicitly articulating the privacy–security tension: detecting poisoned "new knowledge" may require richer auditing/validation signals, but those signals may conflict with client privacy or secure aggregation constraints. We will connect this to our earlier assumption of trusted setups and then explicitly broaden to semi-honest/malicious settings, consistent with Section 2.3’s note that most existing works assume trusted settings, while we discuss threats separately.

---

> ### Author Response · Authors · 2026-01-14
> **Response to Reviewer xne5 [Part 3]**
>
> > Missing important relevant references: Several missing references are listed (healthcare FL, graph FL, security, privacy).
>
> We agree that the specific works you listed are important and currently missing. Thus, we have cited all papers in the proper places in the manuscript and cited them properly.
>
> - An optimization-based attack framework that breaks state-of the-art poisoning defenses in FL was presented in (Yang et al., 2024e).
> - A secret sharing–inspired robust distributed backdoor attack against FL was studied in (Yang et al., 2025).
> - Distributed backdoor attacks on federated graph learning and certified defenses were investigated in (Yang et al., 2024d).
> - GraphFL (Wang et al., 2022a) provided an FL framework for semi-supervised node classification on graph-structured data.
> - An adaptive FL approach to break data silos for Parkinson’s disease diagnosis via privacy-preserving facial expression analysis was proposed in (Pang et al., 2025b).
> - A tri-factor adaptive FL framework for Parkinson’s disease diagnosis via multi-source facial expression analysis was introduced in (Pang et al., 2025a).
> - Recent work further aims to harmonize heterogeneous differential privacy mechanisms and privacy budgets across clients/rounds to improve both accuracy and convergence in DP-FL (Feng et al., 2024a).
> - From a representation perspective, Soteria provides a provable defense by perturbing privacy-sensitive representation components to reduce reconstruction leakage while preserving utility (Sun et al., 2021).
> - Complementarily, task-agnostic privacy-preserving representation learning has been proposed to mitigate attribute inference attacks in FL without assuming a specific downstream task (Arevalo et al., 2024).
> -  In this context, FedTilt studied robust FL under persistent outliers while incorporating multi-level fairness constraints across clients and groups (Zhang et al., 2025b).
> - Provenance primitives such as watermarking can further support ownership verification and model provenance tracking; for example, FedGMark provided a certifiably robust watermarking approach for federated graph learning (Yang et al., 2024c).

---

> > ### Comment · Reviewer_xne5 · 2026-01-18
> >
> > Thanks for the response. The revision addresses my comments, and I lean toward acceptance.

---

### Comment · Reviewer_xne5 · 2025-12-28

**Summary**

This survey paper systematically reviews the challenge of integrating new knowledge—such as new features, tasks, models, and algorithms—into existing Federated Learning (FL) systems to enable sustainable evolution and extended lifespan without discarding prior investments.  The paper positions itself as the first survey to examine FL literature through the lens of system evolution and lifespan extension due to new knowledge integration, differentiating it from prior surveys focused on static settings or isolated techniques.


**Strengths**

+ The focus on FL system evolution and lifespan is timely and addresses a critical gap between rapid ML advancements and the high cost of training FL models from scratch.
+ The 4-category framework (Features, Tasks, Models, Algorithms) is logical and provides a clear structure for reviewing a large body of work.
+ The paper reviews recent literature (up to 2025) across multiple sub-fields within FL.
+ Future work on threats and incentives are valuable for guiding future research.


**Improvement / Clarification**

1. Positioning & Contribution Clarity: The abstract and introduction clearly state the problem. However, the novelty claim as the "first" survey to address FL evolution could be strengthened by more sharply contrasting the proposed evolutionary framework with the "static" assumptions of prior surveys mentioned in Table 1.
2. Structural Flow: Section 3 (New Features) is very detailed and well-organized.  The transition to Section 7 (New Algorithms) feels somewhat abrupt and less developed compared to previous sections. It raises excellent questions but lacks a review of existing work that attempts such transitions. Expanding this section would improve balance.
3. Technical Depth vs. Readability: The draft is highly technical and assumes significant reader expertise in FL. While appropriate for a survey, adding more intuitive explanations of core challenges (e.g., why non-IID data makes FCL so hard) in early sections would improve accessibility.
4. Figures and Visualizations: The referenced figures (Fig 1, 2, 3, 4, 5) are crucial for understanding the taxonomies and workflows. Ensure these are clear, well-labeled, and directly referenced in the text. Their captions should be self-explanatory.
5. Future Directions (Section 8): The discussion on Incentives is insightful, linking contribution evaluation to the knowledge taxonomy. Consider adding a brief diagram or table to summarize the proposed "multi-dimensional model" for quantifying contributions of features, tasks, and models. The Threats subsection rightly identifies new attack surfaces. It could briefly mention the tension between proactive detection of malicious new knowledge (e.g., poisoned features) and the privacy of client data.
6. Missing relevant references:


a) FL for Healthcare

Pang et al.  A Tri-Factor Adaptive Federated Learning Framework for Parkinson's Disease Diagnosis via Multi-Source Facial Expression Analysis. IEEE Journal of Biomedical and Health Informatics, 2025

Pang et al.  Breaking Data Silos in Parkinson’s Disease Diagnosis: An Adaptive Federated Learning Approach for Privacy-Preserving Facial Expression Analysis. AAAI Conference on Artificial Intelligence, 2025

b) FL Algorithm for Graph Data

Wang et al. GraphFL: A Federated Learning Framework for Semi-Supervised Node Classification on Graphs. IEEE International Conference on Data Mining, 2022

Yang et al. FedGMark: Certifiably Robust Watermarking for Federated Graph Learning. Neural Information Processing Systems, 2024


c) Security of FL

Yang et al. A Secret Sharing-Inspired Robust Distributed Backdoor Attack to Federated Learning. ACM Transactions on Privacy and Security, 2025

Zhang et al. FedTilt: Towards Multi-Level Fairness-Preserving and Robust Federated Learning. Deep Learning Security and Privacy Workshop, 2025

Yang et al. Distributed Backdoor Attacks on Federated Graph Learning and Certified Defenses. ACM Conference on Computer and Communications Security, 2024

Yang et al. Breaking State-of-the-Art Poisoning Defenses to Federated Learning: An Optimization-Based Attack Framework. ACM International Conference on Information and Knowledge Management, 2024


d) Privacy of FL

Feng et al.  Universally Harmonizing Differential Privacy Mechanisms for Federated Learning: Boosting Accuracy and Convergence. ACM Conference on Data and Application Security and Privacy, 2025

Arevalo et al. Task-Agnostic Privacy-Preserving Representation Learning for Federated Learning Against Attribute Inference Attacks. AAAI Conference on Artificial Intelligence, 2024

Sun et al. Soteria: Provable Defense against Privacy Leakage in Federated Learning from Representation Perspective. IEEE/CVF Conference on Computer Vision and Pattern Recognition, 2021

**Minors**

1. Some sentences are very long and complex. Breaking them down could enhance readability.

2. Section 2 "Fundamentals" more appropriate as "Preliminaries" or "Setup"

---

### Decision · Action_Editor_NZk6 · 2026-03-03

**Recommendation:** Accept with minor revision

**Additional Comments:**

I noticed the inclusion of FedAlign (Gupta et al., Arxiv 2025) and was a bit confused by it, because there are at least two previous FL algorithms with that name (one of which I co-authored): Ravi et al. (https://openreview.net/forum?id=FR8dvo6q8i) and Ma et al. (https://arxiv.org/abs/2411.15837). Since this is a survey paper, it may be a good place to mention this (unfortunate) overusing of the name and differentiate the methods.

**Audience:**

Yes

**Audience Explanation:**

Members of the TMLR community working on federated learning, distributed/edge ML, continual learning, etc. will find this survey useful.

**Claims And Evidence:**

Yes

**Claims Explanation:**

The paper provides a detailed and systematic review of techniques to integrate new knowledge (e.g., new features, tasks, models, and algorithms) into FL systems. This allows FL systems to be more sustainable and have longer lifespans. This is the first survey of FL literature focused on system evolution and lifespan extension, which differentiates it from prior surveys, and makes it a useful resource for researchers and practitioners in the area.

The reviewers agree that this is a comprehensive survey, covering many papers in the area, and they approve the general taxonomy (new features / tasks / models / algorithms). In the original reviews, some concerns were raised about the objectives for comparison (e.g., the choice of "lifespan" as a core objective). There were also concerns about lack of direct comparisons between the discussed methods. The authors addressed all comments and requested changes, including several new tables comparing the methods across several metrics, and all reviewers were satisfied with the changes.

---

> ### Author Response · Authors · 2026-04-02
> **Camera ready version has been submitted**
>
> Dear Meta Reviewer,
>
> Thank you for your constructive meta review and comments. We have now submitted the camera-ready version of the paper. In this revision, we corrected the typos and made a clearer distinction between the studies that share the name “FedAlign”.
>
> We would be grateful for any further feedback you may have after reviewing the updated version.
>
> Best regards,
>
> Authors of Paper6400